# Enabling Automatic Differentiation
# with Mollified Graph Neural Operators

**Ryan Y. Lin**[1], **Julius Berner**[2*], **Valentin Duruisseaux**[1], **David Pitt**[1], **Daniel Leibovici**[2],
**Jean Kossaifi**[2], **Kamyar Azizzadenesheli**[2], **Anima Anandkumar**[1]

[1]*Caltech*    [2]*NVIDIA*

**Reviewed on OpenReview:** *https://openreview.net/forum?id=CGoR1hFAGr*

## Abstract

Physics-informed neural operators offer a powerful framework for learning solution operators of partial differential equations (PDEs) by combining data and physics losses. However, these physics losses require the efficient and accurate computation of derivatives. Computing these derivatives remains challenging, with spectral and finite difference methods introducing approximation errors due to finite resolution. Here, we propose the mollified graph neural operator ($m$GNO), the first method to leverage automatic differentiation and compute *exact* gradients on arbitrary geometries. This enhancement enables efficient training on arbitrary point clouds and irregular grids with varying geometries while allowing the seamless evaluation of physics losses at randomly sampled points for improved generalization. For a PDE example on regular grids, $m$GNO paired with Autograd reduced the L2 relative data error by 20× compared to finite differences, suggesting it better captures the physics underlying the data. It can also solve PDEs on unstructured point clouds seamlessly, using physics losses only, at resolutions vastly lower than those needed for finite differences to be accurate enough. On these unstructured point clouds, $m$GNO leads to errors that are consistently 2 orders of magnitude lower than machine learning baselines (Meta-PDE, which accelerates PINNs) for comparable runtimes, and also delivers speedups from 1 to 3 orders of magnitude compared to the numerical solver for similar accuracy. $m$GNOs can also be used to solve inverse design and shape optimization problems on complex geometries.

## 1 Introduction

PDEs are critical for modeling physical phenomena relevant to scientific applications. Unfortunately, numerical solvers become very expensive computationally when used to simulate large-scale systems. To avoid these limitations, neural operators, a machine learning paradigm, have been proposed to learn solution operators of PDEs (Azizzadenesheli et al., 2024; Berner et al., 2025). Neural operators learn mappings between function spaces rather than finite-dimensional vector spaces, and, in particular can be used to approximate solution operators of PDE families (Li et al., 2020b; Kovachki et al., 2023). One example is the Fourier Neural Operator (FNO) (Li et al., 2020a), which relies on Fourier integral transforms with kernels parameterized by neural networks. Another example, the Graph Neural Operator (GNO) (Li et al., 2020b), implements kernel integration with graph structures and is applicable to complex geometries and irregular grids. The GNO has been combined with FNOs in the Geometry-Informed Neural Operator (GINO) (Li et al., 2023) to handle arbitrary geometries when solving PDEs. Neural operators have been successfully used to solve PDE problems with significant speedups (Kurth et al., 2022; Gopakumar et al., 2023; Wen et al., 2023; Ghafourpour et al., 2025). They have shown great promise, primarily due to their ability to receive input functions at arbitrary discretizations and query output functions at arbitrary points, and also due to their universal operator approximation property (Kovachki et al., 2021).

---

*Work done partially at Caltech.

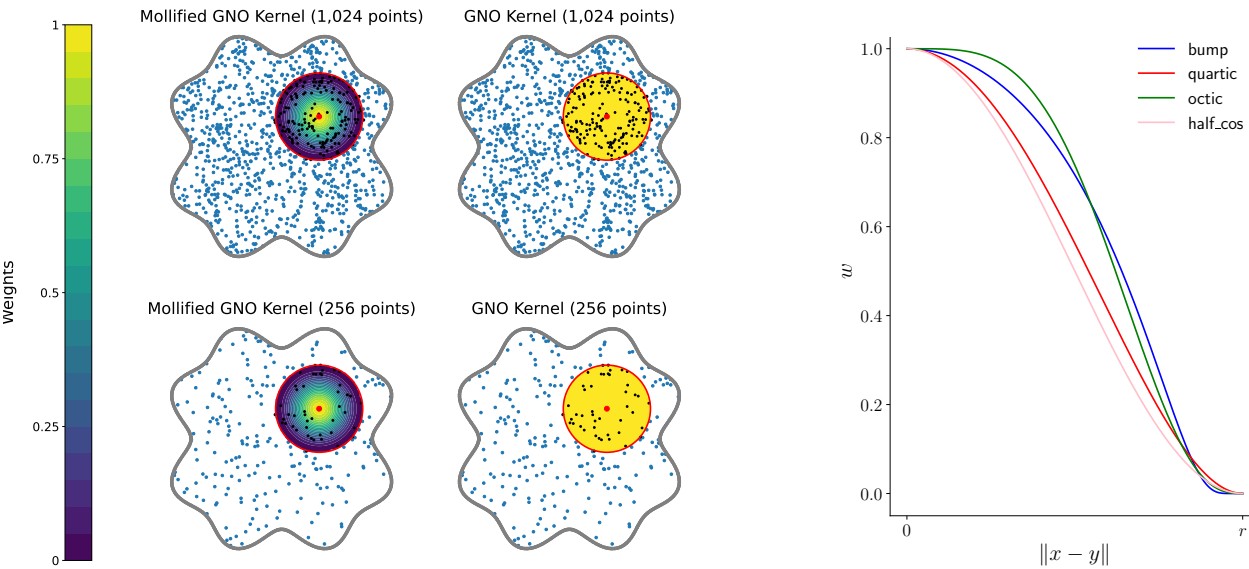

Figure 1: *(left)* Mollified GNO kernel's neighborhood and weights versus those of the vanilla GNO kernel with differing point densities. *(right)* Examples of weight functions (equation 9 - equation 12) for $m$GNO.

However, purely data-driven approaches may underperform in situations with limited or low-resolution data (Li et al., 2021b), and may be supplemented using knowledge of physics laws (Karniadakis et al., 2021) as additional loss terms, as done in Physics-Informed Neural Networks (PINNs) (Raissi et al., 2017a;b; 2019). In the context of neural operators, the Physics-Informed Neural Operator (PINO) (Li et al., 2021b) combines training data (when available) with a PDE loss at a higher resolution, and can be fine-tuned on a given PDE instance using only the equation loss to provide a nearly zero PDE error at all resolutions.

A major challenge when using physics losses is to efficiently compute derivatives without sacrificing accuracy, since numerical errors on the derivatives are compounded in the physics losses. Approximate derivatives can be computed using finite differences, but can require a very high resolution grid to be sufficiently accurate, thus becoming intractable for fast-varying dynamics. Numerical derivatives can also be computed using Fourier differentiation, but this requires smoothness, uniform grids, and performs best when applied to periodic problems. In contrast, automatic differentiation computes exact derivatives using repeated applications of the chain rule and scales better to large-scale problems and fast-varying dynamics. Unlike numerical differentiation methods, which introduce errors, automatic differentiation gives exact gradients, ensuring the accuracy required for physics losses, making it the preferred approach for physics-informed machine learning.

**Approach.** We propose a fully differentiable modification to GNOs to allow for the use of automatic differentiation when computing derivatives and physics losses, which was previously prohibited by the GNO's non-differentiability. More precisely, we replace the non-differentiable indicator function in the GNO kernel integration by a differentiable weighting function. This is inspired by mollifiers in functional analysis (Evans, 2010), which are used to approximate, regularize, or smooth functions.

As a result, our differentiable mollified GNO ($m$GNO) is the first method capable of computing exact derivatives at arbitrary query points. The resulting $m$GNO can, in particular, be used within GINO to learn efficiently and accurately the solution operator of families of large-scale PDEs with varying geometries without data using physics losses. A data loss can help further improve the results when reference simulations are available, even if these are of very low resolution, when paired with a high-resolution physics loss.

**Summary of Results.** We test our approach on Burgers' and Navier—Stokes equations with regular grids, and on nonlinear Poisson and hyperelasticity equations with varying domain geometries. Figure 2 shows example solutions, highlighting the complexity of the geometries considered. Physics losses prove critical, emphasizing the need for accurate and efficient derivative computation.

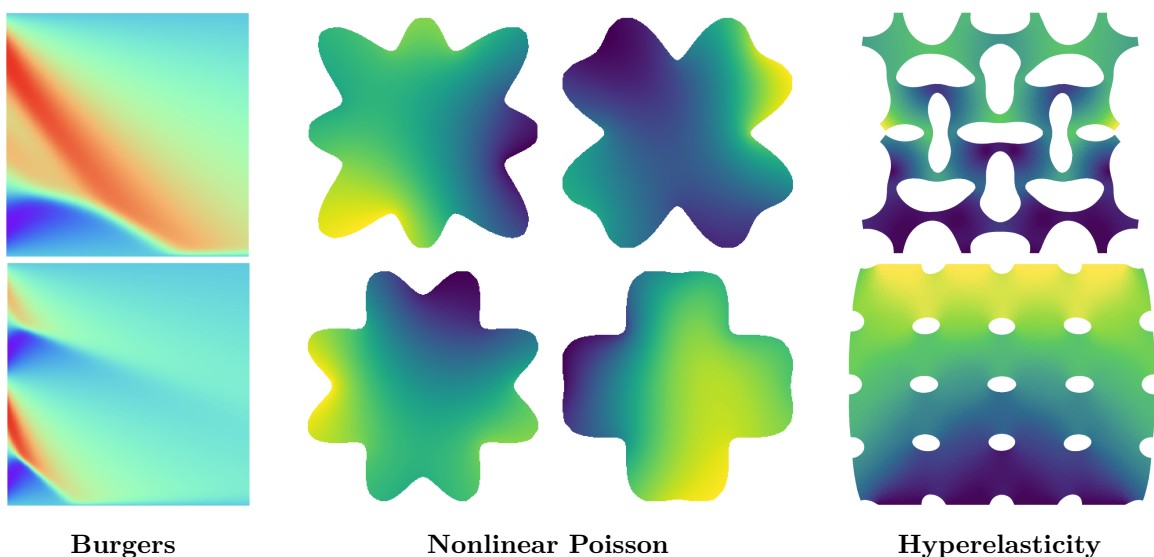

Figure 2: Examples of solutions for the problems considered.

Using Autograd instead of finite differences reduces the relative L2 data loss by 20× for Burgers' equation on regular grids, showing that the Autograd physics loss captures the underlying physics more accurately. In hybrid training with noisy reference data, the Autograd-based $m$GNO ∘ FNO remains robust and accurate, maintaining low PDE residuals and data loss.

It also excels at learning the lid cavity flow example governed by the Navier–Stokes equations. For this problem, access to data is essential, but a small number of randomly sampled points (with Autograd) at each iteration achieves excellent results when trained with a hybrid loss. Including a physics loss is also critical, as it ensures PDE fidelity and guides the network toward physically consistent solutions even at low resolution.

Autograd $m$GINO performs seamlessly on unstructured point clouds, while finite difference derivatives are not sufficiently accurate at the same training resolution and would need at least 9× more points. Autograd $m$GINO achieves a relative error 2-3 orders of magnitude lower than the machine learning baselines (Meta-PDE (Qin et al., 2022) using LEAP and MAML) considered for a comparable running time, and enjoys speedups of 20-25× and 3000-4000× compared to the numerical solver for similar accuracy on the Poisson and hyperelasticity equations. Furthermore, as a result of its differentiability, $m$GINO can be used seamlessly for solving inverse design and shape optimization problems on complex geometries, as demonstrated with an airfoil design problem.

## 2 Background

### 2.1 Neural Operators

Neural operators compose linear integral operators $\mathcal{K}$ with pointwise nonlinear activation functions $\sigma$ to approximate nonlinear operators. A **neural operator** is defined as

$$\mathcal{Q} \circ \sigma(W_L + \mathcal{K}_L + b_L) \circ \cdots \circ \sigma(W_1 + \mathcal{K}_1 + b_1) \circ \mathcal{P} \tag{1}$$

where $\mathcal{Q}$ and $\mathcal{P}$ are the pointwise neural networks that encode (lift) the lower dimension function onto a higher-dimensional space and project it back to the original space, respectively. The model stacks $L$ layers of $\sigma(W_l + \mathcal{K}_l + b_l)$ where $W_l$ are pointwise linear operators (matrices), $\mathcal{K}_l$ are integral kernel operators, $b_l$ are bias terms, and $\sigma$ are fixed activation functions.

The parameters of a neural operator consist of all the parameters in $\mathcal{P}, \mathcal{Q}, W_l, \mathcal{K}_l, b_l$. Kossaifi et al. (2025) maintains a comprehensive open-source library for learning neural operators in PyTorch, which serves as the foundation for our implementation.

**Fourier Neural Operator (FNO).** A FNO (Li et al., 2020a) is a neural operator using *Fourier integral operator* layers

$$\big(\mathcal{K}(\phi)v_t\big)(x) = \mathcal{F}^{-1}\Big(R_\phi \cdot (\mathcal{F}v_t)\Big)(x) \tag{2}$$

where $R_\phi$ is the Fourier transform of a periodic function $\kappa$ parameterized by $\phi$. When the discretizations of both the input and output functions are given on a uniform mesh, the Fourier transform $\mathcal{F}$ can be implemented using the fast Fourier transform. We note that the FNO architecture is fully differentiable. A depiction of the FNO architecture is provided in Figure 14.

**Graph Neural Operator (GNO).** A GNO (Li et al., 2020b) implements kernel integration with graph structures and is applicable to complex geometries and irregular grids. The GNO kernel integration shares similarities with the message-passing implementation of graph neural networks (GNNs) (Battaglia et al., 2016). However, GNO defines the graph connection in a ball in continuous physical space, while GNN assumes a fixed set of neighbors in a discrete graph. The GNN nearest-neighbor connectivity violates discretization convergence and degenerates into a pointwise operator at high resolutions, leading to a poor approximation of the operator. In contrast, GNO adapts the graph based on points within a physical space, allowing for universal approximation of operators.

Specifically, the GNO acts on an input function $v$ as follows,

$$\mathcal{G}_{\mathrm{GNO}}(v)(x) \coloneqq \int_D \mathbb{1}_{\mathrm{B}_r(x)}(y)\kappa(x,y)v(y)\,\mathrm{d}y, \tag{3}$$

where $D \subset \mathbb{R}^d$ is the domain of $v$, $\kappa$ is a learnable kernel function, and $\mathbb{1}_{\mathrm{B}_r(x)}$ is the indicator function over the ball $\mathrm{B}_r(x)$ of radius $r > 0$ centered at $x \in D$. The radius $r$ is a hyperparameter, and the integral can be approximated with a Riemann sum, for instance.

**Geometry-Informed Neural Operator (GINO).** GINO (Li et al., 2023) proposes to combine an FNO with GNOs to handle arbitrary geometries. More precisely, the input is passed through three main neural operators,

$$\mathcal{G}_{\mathrm{GINO}} = \mathcal{G}_{\mathrm{GNO}}^{decoder} \circ \mathcal{G}_{\mathrm{FNO}} \circ \mathcal{G}_{\mathrm{GNO}}^{encoder}. \tag{4}$$

First, a GNO encodes the input given on an arbitrary geometry into a latent space with a regular geometry. The encoded input can be concatenated with a signed distance function evaluated on the same grid if available. Then, an FNO is used as a mapping on that latent space for efficient global integration. Finally, a GNO decodes the output of the FNO by projecting that latent space representation to the output geometry. As a result, GINO can represent mapping from one complex geometry to another. A depiction of the GINO architecture is displayed in Figure 15.

## 2.2 Physics-Informed Machine Learning

**Physics-Informed Neural Network (PINN).** In the context of solving PDEs, a PINN (Raissi et al., 2017a;b; 2019) is a neural network representation of the solution of a PDE, whose parameters are learned by minimizing the distance to the reference PDE solution and deviations from known physics laws such as conservation laws, symmetries and structural properties, and from the governing differential equations. PINNs minimize a composite loss

$$\mathcal{L} = \mathcal{L}_{\mathrm{data}} + \lambda\mathcal{L}_{\mathrm{physics}} \tag{5}$$

where $\mathcal{L}_{\mathrm{data}}$ measures the error between data and model predictions while $\mathcal{L}_{\mathrm{physics}}$ penalizes deviations away from physics laws. PINN overcomes the need to choose a discretization grid of most numerical solvers, but only learns the solution for a single PDE instance and does not generalize to other instances without further optimization. Many modified versions of PINNs have also been proposed and used to solve PDEs in numerous contexts (Jagtap et al., 2020; Cai et al., 2022; Yu et al., 2022). When data is not available, we can try to learn the solution by minimizing $\mathcal{L}_{\mathrm{physics}}$ only.

While physics losses can prove very useful, the resulting optimization task can be challenging and prone to numerical issues. The training loss typically has poor conditioning as it involves differential operators that can be ill-conditioned. In particular, Krishnapriyan et al. (2021) showed that the loss landscape becomes increasingly complex and harder to optimize as the physics loss coefficient $\lambda$ increases. The model could also converge to a trivial or non-desired solution that satisfies the physics laws on the set of points where the physics loss is computed (Leiteritz & Pflüger, 2021). There could also be conflicts between the multiple loss terms when their gradients point in opposite directions. Even without such conflicts, the losses can vary significantly in magnitude, leading to unbalanced backpropagated gradients during training (Wang et al., 2021), and thus to the contribution and reduction of certain losses being relatively negligible during training. Manually tuning the loss coefficients can be computationally expensive, especially as the number of loss terms increases. Note that strategies have been developed to adaptively update the loss coefficients and mitigate this issue (Chen et al., 2018; Heydari et al., 2019; Bischof & Kraus, 2021).

**Physics-Informed Neural Operator (PINO).** In the *physics-informed neural operator* approach (Li et al., 2021b), an FNO is trained with low-resolution simulations (or even without data) and high-resolution physics losses, allowing for near-perfect approximations of PDE solution operators. To further improve accuracy at test time, trained models can be fine-tuned on a given PDE instance using only the equation loss and provide a near-zero error at all resolutions. PINO has been successfully applied to many PDEs (Song et al., 2022; Meng et al., 2023; Rosofsky et al., 2023). In practice, the PDE loss vastly improves generalization, physical validity, and data efficiency in operator learning compared to purely data-driven methods.

**Physics-informed approaches without physics losses.** Various approaches enforce physics laws in surrogate models without using physics losses. This can be achieved, for instance, by using projection layers (Jiang et al., 2020; Duruisseaux et al., 2024; Harder et al., 2024), by finding optimal linear combinations of learned basis functions that solve a PDE-constrained optimization problem (Negiar et al., 2022; Chalapathi et al., 2024), by leveraging known characterizations and properties of the solution operator as for divergence-free flows (Richter-Powell et al., 2022; Mohan et al., 2023; Xing et al., 2024), or Hamiltonian systems with their symplectic structure (Burby et al., 2020; Jin et al., 2020; Chen & Tao, 2021; Duruisseaux et al., 2023). These approaches are particularly advantageous when the system is well-understood and physical constraints can be explicitly encoded, ensuring strict compliance with the underlying physics. They also tend to require less computational overhead than methods that rely on minimizing physics losses. However, such approaches are often tailored to specific PDE structures and are limited to dynamical systems with well-characterized solutions. In contrast, physics-loss-based methods offer greater flexibility and broader applicability across a wide range of PDEs, including systems whose solution properties are not fully understood. By incorporating physics-based loss terms during training, as in PINNs and PINOs, the governing equations serve as regularizers, offering greater flexibility, broad applicability across diverse PDEs, and the advantage of requiring only knowledge of the underlying equations. Hybrid strategies are also possible, in which certain constraints are enforced directly in the architecture while others are imposed through physics losses during training. For instance, in incompressible Navier–Stokes problems, a physics loss can penalize deviations from the momentum equation, while the divergence-free condition can be enforced explicitly in the model (e.g. using projections as done in Jiang et al. (2020) and Duruisseaux et al. (2024)).

**Physics-Infused Transformer Architectures.** Recent parallel research has explored embedding certain structures of PDEs into Transformer architectures. Transolver (Wu et al., 2024; Luo et al., 2025) introduces a physics-attention mechanism that adaptively groups mesh points with similar physical states, enabling the model to capture complex correlations across arbitrary geometries efficiently. Unisolver (Zhou et al., 2025) conditions a Transformer on complete physics information, incorporating both domain-wise components, such as equation symbols and coefficients, and point-wise components such as boundary conditions, achieving broad applicability across diverse PDEs. PDEFormer (Ye et al., 2025) represents PDEs as computational graphs and combines symbolic and numerical information through a graph Transformer and implicit neural representation, allowing mesh-free solutions and efficient handling of varied PDE types. Collectively, these architectures illustrate an alternative approach to integrating physical knowledge into Transformer structures, offering complementary strategies for improving the accuracy and flexibility of machine learning surrogates for PDE solvers.

## 2.3 Computing Derivatives

To use physics losses, a major technical challenge is to efficiently compute derivatives without sacrificing accuracy, since numerical errors on the derivatives will be amplified in the physics losses and output solution.

**Finite Differences.** A simple approach is to use numerical derivatives computed using finite differences. This differentiation method is fast and memory-efficient (it requires $O(n)$ computations for an $n$-point grid). However, it faces the same challenges as finite difference numerical solvers: it requires a fine-resolution grid to be accurate and therefore becomes intractable for multi-scale and fast-varying dynamics. On point clouds, the stencil coefficients in finite difference formulas vary from point to point and must be computed each time, as outlined in Appendix E, adding to the computational cost. The errors in the resulting derivatives will also vary across the domain depending on the density of nearby points.

**Fourier Differentiation.** Fourier differentiation is also fast and memory-efficient to approximate derivatives as it requires $O(n \log n)$ given an $n$-point grid. However, just like spectral solvers, it requires smoothness, uniform grids, and performs best when applied to periodic problems. Fourier differentiation can be performed on non-uniform grids, but the computational cost grows to $O(n^2)$, and if the target function is non-periodic or non-smooth, the Fourier differentiation is not accurate. To deal with this issue, the Fourier continuation method (Ganeshram et al., 2025) can be applied to embed the problem domain into a larger periodic space, at the cost of higher computational and memory complexity.

**Pointwise Differentiation with Autograd.** Derivatives can be computed pointwise using automatic differentiation by applying the chain rule to the sequence of operations in the model. Autograd (Maclaurin et al., 2015) automates this process by constructing a computational graph during the forward pass and leveraging reverse-mode differentiation to compute gradients during the backward pass. Autograd is typically the preferred method for computing derivatives in PINNs for a variety of reasons (Baydin et al., 2017):

- Unlike finite differences, which can introduce large discretization errors when not computed on fine enough grids, Autograd provides *exact* derivatives, ensuring the accuracy that is critical in physics-informed machine learning, where derivative errors are amplified in physics losses. By *exact* derivatives, we mean that automatic differentiation computes gradients precisely from the differentiable computation graph, accurate up to machine precision. While the graph represents a discretized approximation of the continuous PDE, in practice, the computed loss and gradients are extremely accurate, and automatic differentiation remains highly reliable even when evaluated on a relatively small set of points, as demonstrated in our numerical experiments.

- Autograd computes gradients in a single pass (while finite differences require multiple function evaluations). It provides exact derivatives regardless of the mesh resolution, making it particularly advantageous for large-scale problems, where finite differences become computationally intractable.

- Autograd can compute higher-order derivatives with minimal additional cost, while finite differences require additional function evaluations and can suffer from further error accumulation.

- Autograd can compute derivatives at any point in the domain seamlessly, while finite differences can struggle to handle complex geometries.

- Figure 3 empirically shows that Autograd derivatives are smoother and more stable than finite differences with the proposed model.

However, Autograd also has limitations: all operations need to be differentiable, and storing all intermediate computations in a computational graph can significantly increase memory usage for deep models. As a result, it can be slower and more memory-intensive than finite differences for simpler low-dimensional problems for which finite differences are accurate enough at low resolutions. When the physics loss involves a deep composition of operations, issues of vanishing or exploding gradients can also be exacerbated during backward propagation with automatic differentiation.

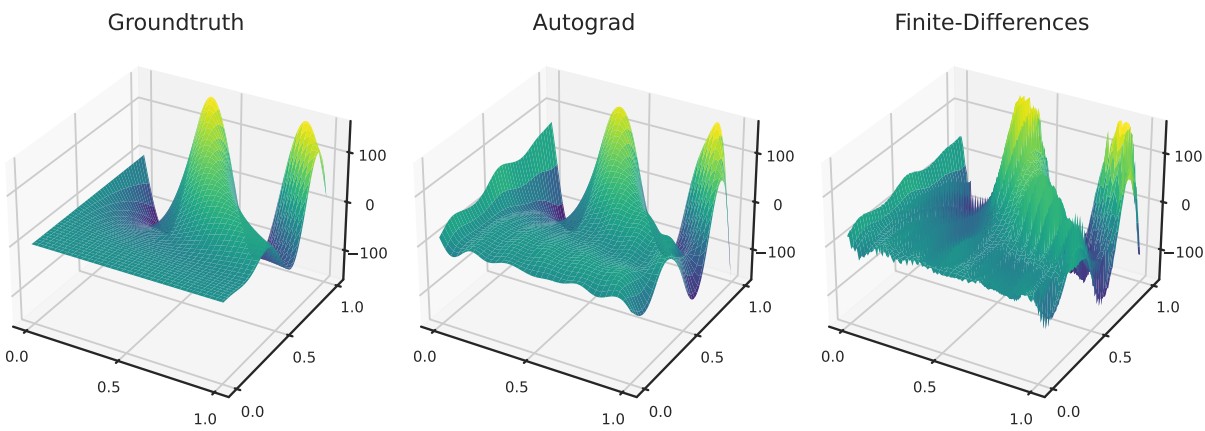

Figure 3: Qualitative assessment of the smoothness of derivatives. We fit $m$GINO with a physics loss to the differentiable function $u(x,y) = \sin(4\pi xy)$ on $[0,1]^2$, and compare the analytic ground truth derivative $\partial_{xx}u(x,y) = -16\pi^2 y^2 \sin(4\pi xy)$ (*left*) with the numerical derivatives obtained using automatic differentiation (*middle*) and finite differences (*right*).

## 3 Methodology

**Mollified GNO.** Recall from Equation (3) that the GNO computes the integral of an indicator function $\mathbb{1}_{\mathrm{B}_r(x)}$ that is not differentiable. We propose a fully differentiable layer that replaces the indicator function $\mathbb{1}_{\mathrm{B}_r(x)}$ with a differentiable weight function $w$ supported within $\mathrm{B}_r(x)$. This is inspired by the mollifier in functional analysis (Evans, 2010), which is used to smooth out the indicator function. The resulting *mollified graph neural operator* ($m$GNO) acts on an input function $v$ via

$$\mathcal{G}_{m\mathrm{GNO}}(v)(x) \coloneqq \int_{\widetilde{D}} w(x,y)\kappa(x,y)v(y)\,\mathrm{d}y, \tag{6}$$

for $x \in D$. Here, we padded the input of the FNO such that its output function $v$ is supported on the extended domain

$$\widetilde{D} \coloneqq D + \mathrm{B}_r(0) = \{x + y \mid x \in D,\ y \in \mathrm{B}_r(0)\},$$

which allows the recovery of exact derivatives at the boundary. Then, we can compute the derivative

$$\partial_x\, \mathcal{G}_{m\mathrm{GNO}}(v)(x) = \int_{\mathrm{B}_r(x)} \partial_x[w(x,y)\kappa(x,y)]v(y)\,\mathrm{d}y. \tag{7}$$

Automatic differentiation algorithms can then be used to compute the derivatives appearing in physics losses. We can also use a cached version of neighbor search (i.e. store neighbors with nonzero weight) to keep the method as efficient as the original GNO up to the negligible cost of evaluating the weight function $w$. We emphasize that these definitions work for arbitrarily complex domains $D$. As for the GNO, the radius $r$ is a hyperparameter and the integral can be approximated with a Riemann sum, for instance. An example of simplified pseudocode for the $m$GNO layer is provided in Appendix D.

**Mollified GINO.** The mollified GNO can then be used within a fully differentiable *mollified geometry-informed neural operator* ($m$GINO) to learn efficiently the solution operator of PDEs with varying geometries using physics losses where the derivatives are computed using automatic differentiation,

$$\mathcal{G}_{m\mathrm{GINO}} = \mathcal{G}_{m\mathrm{GNO}}^{decoder} \circ \mathcal{G}_{\mathrm{FNO}} \circ \mathcal{G}_{m\mathrm{GNO}}^{encoder}. \tag{8}$$

This allows $m$GINO to be used for solving inverse design and shape optimization problems on complex geometries, by backpropagating gradients with respect to control parameters through it.

**Weight Functions.** The choice of weight function plays a central role in mollifying the hard cutoff neighborhoods of standard GNOs with smooth, compactly supported functions that enable stable and accurate automatic differentiation. These weights gradually attenuate contributions near the neighborhood boundary, improving gradient stability and reducing artifacts. Key properties include smoothness to avoid spurious gradient oscillations, compact support to control computational cost by ignoring distant points, and locality tuned through the radius $r$.

In practice, smooth bell-shaped or cosine-like profiles often perform well and are well-suited. The radius $r$ is typically chosen to obtain a desired typical number of neighbors per query point which balances efficiency and information aggregation. Performance is poor when the radius is too small, and when $r$ becomes too large, the computational and memory costs significantly increase while performance deteriorates. In practice, one can start from a smaller value of $r$ and progressively increase it until satisfactory performance is achieved.

Letting $d = \|x - y\|_2 / r$, examples of weight functions $w$ with support in $\mathbb{1}_{B_r(x)}(y)$ are

$$w_{\text{bump}}(x, y) \coloneqq \mathbb{1}_{B_r(x)}(y) \ \exp\big(d^2/(d^2 - 1)\big), \tag{9}$$

$$w_{\text{quartic}}(x, y) \coloneqq \mathbb{1}_{B_r(x)}(y) \left(1 - 2d^2 + d^4\right), \tag{10}$$

$$w_{\text{octic}}(x, y) \coloneqq \mathbb{1}_{B_r(x)}(y) \left(1 - 6d^4 + 8d^6 - 3d^8\right), \tag{11}$$

$$w_{\text{half\_cos}}(x, y) \coloneqq \mathbb{1}_{B_r(x)}(y) \left[0.5 + 0.5\cos(\pi d)\right]. \tag{12}$$

These weight functions, displayed on the right in Figure 1, are decreasing functions from 1 to 0 on $[0, r]$, and preserve differentiability for all points.

An ablation study for the radius $r$ and choice of weight function $w$ is conducted for the nonlinear Poisson equation in Section 4.2.6, and a discussion on the choice of aggregation scheme for neighbors contributions is provided in Section 4.2.3, in particular to highlight how a *mean* aggregation can lead to undesired artifacts when training based on a data loss only.

**Remark.** The entire $m$GNO and $m$GINO architectures are designed such that all of their components are differentiable, ensuring that derivatives of any order can be computed correctly and accurately using automatic differentiation. For instance, we avoid non-differentiable activation functions such as ReLU (Rectified Linear Unit), which introduce discontinuities in their derivatives, and instead use smooth alternatives like GeLU (Gaussian Error Linear Unit). All the other operations in the network, including the Fourier layers, pointwise linear neural networks, and pointwise linear operators, are also implemented in a differentiable manner. This design eliminates issues related to subgradients or undefined derivatives at specific points and preserves the validity of higher-order derivative computations using automatic differentiation. A discussion of using Autograd with non-differentiable components is included in Appendix G.

## 4 Numerical Experiments

We use Meta-PDE (Qin et al., 2022) and the popular finite element method (FEM) FEniCS (Alnæs et al., 2015; Logg et al., 2011) as baselines. Meta-PDE learns initializations for PINNs over multiple instances that can be fine-tuned on any single instance, and two versions have been proposed based on the meta-learning algorithms MAML (Finn et al., 2017) and LEAP (Flennerhag et al., 2019). While FEniCS has been successfully applied in various disciplines, it has not been highly optimized for our applications and could possibly be outperformed by other FEMs and non-FEMs (Liu, 2009) such as Radial Basis Function (RBF) methods (interpolation using RBFs), Finite Point Methods (FPMs) and Moving Least Squares (MLS) methods (weighted least squares to approximate solutions) and spectral methods (expanding the solution in a basis of functions).

### 4.1 Burgers' Equation

#### 4.1.1 Problem Description

The Burgers' equation models the propagation of shock waves and the effects of viscosity in fluid dynamics. We consider the 1D time-dependent Burgers' equation with periodic boundary conditions, and initial condition $u_0 \in L^2_{\text{per}}(D; \mathbb{R})$ with $D = (0, 1)$. The goal is to learn the mapping $\mathcal{G}^{\dagger}$ from the initial condition $u(x, 0) = u_0$ to the solution $u(x, t)$ of the following differential equation for $x \in D$:

$$\partial_t u(x, t) + \partial_x(u^2(x, t)/2) = \nu \partial_{xx} u(x, t), \qquad t \in (0, 1]$$
$$u(x, 0) = u_0(x).$$

We focus on the dataset of Li et al. (2021a;b) consisting of 800 instances of the Burgers' equation with viscosity coefficient $\nu = 0.01$ and $128 \times 26$ resolution. Each sample in the dataset corresponds to a different initial condition $u_0$ drawn from the Gaussian process $\mathcal{N}(0, 625(-\Delta + 25I)^{-2})$. Examples of initial conditions and solutions to this time-dependent Burgers' equation are shown in Figures 4 and 5.

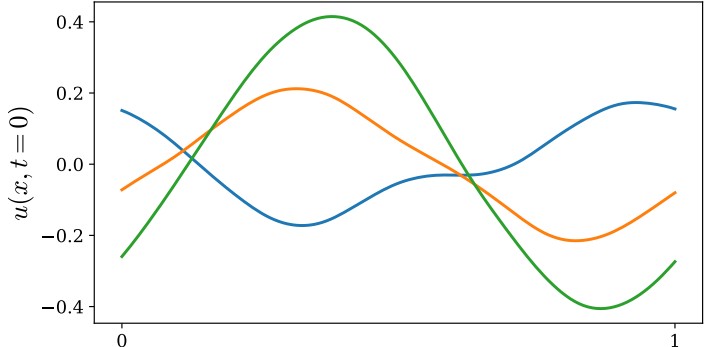

Figure 4: Examples of initial conditions used for the Burgers' Equation.

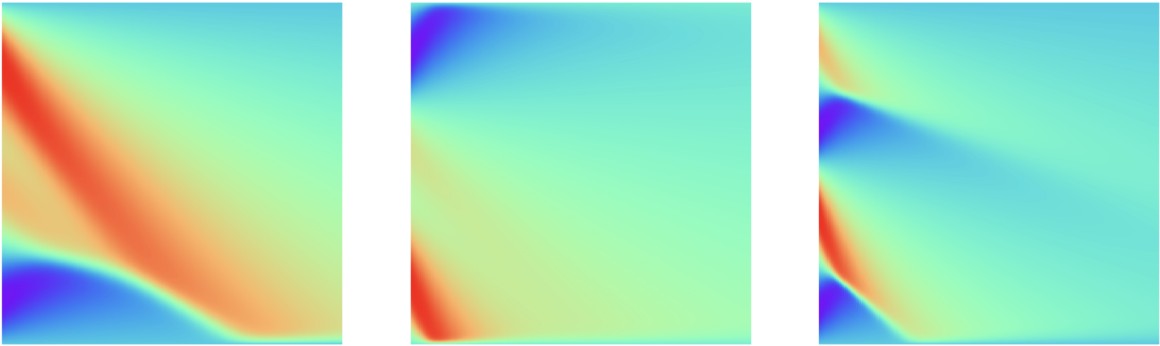

Figure 5: Ground truth solutions for several initial conditions of the Burgers' Equation

The input initial condition $u_0(x)$ given on a regular spatial grid is first duplicated along the temporal dimension to obtain a 2D regular grid, and then passed through a 2D FNO and a mollified GNO to produce a predicted function

$$v = (\mathcal{G}_{m\text{GNO}} \circ \mathcal{G}_{\text{FNO}})(u_0) \qquad (13)$$

approximating the solution $u(x, t)$. We minimize a weighted sum of the PDE residual and initial condition loss,

$$\mathcal{L}_{\text{Burgers}}(v) = \left\| \partial_t v + \partial_x(v^2/2) - \nu \partial_{xx} v \right\|^2_{L^2(D \times (0,1))} + \alpha \left\| v(\cdot, 0) - u_0 \right\|^2_{L^2(D)}. \qquad (14)$$

### 4.1.2 Comparison of Differentiation Methods and Training Strategies

We trained models using $\mathcal{L}_{\mathrm{Burgers}}$, where the derivatives are computed using various differentiation methods. The results, obtained by fine-tuning the models one instance at a time and then averaging over the dataset, are presented in Table 1. Compared to numerical differentiation, $m\mathrm{GNO} \circ \mathrm{FNO}$ with Autograd was easier to tune and produced a model with data loss $20\times$ lower (although $6\times$ slower per training epoch), suggesting it better captures the physics underlying the data. It also achieved a physics loss $3000\times$ smaller, but this was expected since the physics loss is evaluated using Autograd.

Table 1: Data and physics losses (computed using Autograd) for the proposed $m\mathrm{GNO} \circ \mathrm{FNO}$ models and PINO baselines, trained with $\mathcal{L}_{\mathrm{Burgers}}$ where derivatives are computed using different methods.

|  | PDE Residual Loss | Relative L2 Data Loss |
|---|---|---|
| Autograd (ours) | $1.62 \cdot 10^{-6}$ | $1.33 \cdot 10^{-2}$ |
| Finite Differences | $4.89 \cdot 10^{-3}$ | $2.24 \cdot 10^{-1}$ |
| Fourier Differentiation | $4.77 \cdot 10^{-3}$ | $2.19 \cdot 10^{-1}$ |
| PINO Baselines: |  |  |
|   Autograd | $1.87 \cdot 10^{-9}$ | $1.22 \cdot 10^{-1}$ |
|   Finite Differences | $3.71 \cdot 10^{-4}$ | $2.26 \cdot 10^{-1}$ |
|   Fourier Differentiation | $3.63 \cdot 10^{-4}$ | $2.19 \cdot 10^{-1}$ |
| Transolver Baseline (finite-differences) | $3.01 \cdot 10^{-2}$ | $7.47 \cdot 10^{-2}$ |

A comparison to PINO (i.e. using an FNO instead of a $m\mathrm{GNO} \circ \mathrm{FNO}$) shows that PINO achieved a lower physics loss, but failed to reduce the Relative L2 data loss below 10% on the hyperparameter sweep considered. Note that Autograd $m\mathrm{GNO} \circ \mathrm{FNO}$ was only $1.2\times$ slower than Autograd PINO (although this was with a batch size of 1 and Autograd PINO could be accelerated more easily using batching). We also compared performance to Transolver (Wu et al., 2024) with finite differences. We performed a grid search over learning rate, hidden dimension, number of heads and layers, initial condition loss coefficient, and reported the best result in Table 1. Transolver does not reduce the PDE residual as effectively as the other approaches, but strikes a better balance by achieving lower data loss than the remaining baselines. Nonetheless, it still falls short of the performance of Autograd $m\mathrm{GNO} \circ \mathrm{FNO}$, which attains a substantially lower PDE residual while staying closer to the reference data.

We trained $m\mathrm{GNO} \circ \mathrm{FNO}$ models using a data loss, a physics loss, and a hybrid loss (data and physics). The results displayed in Table 2 show that using the data loss helps reduce the gap with reference data, but can come at the cost of a higher physics loss, and vice versa.

Table 2: Data and PDE loss of three models trained with different losses $\mathcal{L}$, using Autograd to compute the physics losses, for the Burgers' equation.

|  | $\mathcal{L}_{\mathrm{data}}$ | $\mathcal{L}_{\mathrm{data}} + \lambda\mathcal{L}_{\mathrm{physics}}$ | $\mathcal{L}_{\mathrm{physics}}$ |
|---|---|---|---|
| Relative L2 Data Loss | $1.42 \cdot 10^{-3}$ | $3.81 \cdot 10^{-3}$ | $1.33 \cdot 10^{-2}$ |
| PDE Residual Loss | $4.80 \cdot 10^{-4}$ | $1.48 \cdot 10^{-4}$ | $1.62 \cdot 10^{-6}$ |

We also tried pre-training the $m\mathrm{GNO} \circ \mathrm{FNO}$ models using the data loss before fine-tuning them with the physics loss $\mathcal{L}_{\mathrm{Burgers}}$ only, but were not able to obtain better results this way. The data loss rapidly deteriorates from its original value obtained in a data-driven way, while the physics loss improves slowly but does not get better than when training with hybrid and physics-only losses without data loss pre-training.

### 4.1.3 Subsampling and Data Efficiency

We now consider randomly subsampling the points at which the PDE residuals are evaluated to consider the effect of reducing the resolution used for the physics loss on performance. Table 3 shows that despite the random locations and reduction of the number of points used to evaluate the derivatives, we maintain good results until the number of points becomes very low.

Table 3: PDE Residual and relative data losses for $m\mathrm{GNO} \circ \mathrm{FNO}$ models trained with the physics loss $\mathcal{L}_{\mathrm{Burgers}}$ using Autograd derivatives evaluated at different numbers of randomly subsampled points from the original $128 \times 26$ regular grid with 3328 points.

|  | PDE Residual Loss | Relative L2 Data Loss |
|---|---|---|
| Full Grid (3328 points) | $1.62 \cdot 10^{-6}$ | $1.33 \cdot 10^{-2}$ |
| Random 1000 points | $6.70 \cdot 10^{-6}$ | $1.32 \cdot 10^{-2}$ |
| Random 500 points | $8.15 \cdot 10^{-6}$ | $1.33 \cdot 10^{-2}$ |
| Random 250 points | $1.47 \cdot 10^{-5}$ | $1.44 \cdot 10^{-2}$ |
| Random 100 points | $4.06 \cdot 10^{-5}$ | $1.91 \cdot 10^{-2}$ |
| Random 50 points | $5.31 \cdot 10^{-5}$ | $2.19 \cdot 10^{-2}$ |
| Random 25 points | $1.29 \cdot 10^{-4}$ | $2.93 \cdot 10^{-2}$ |
| Random 10 points | $3.49 \cdot 10^{-4}$ | $4.62 \cdot 10^{-2}$ |

We also investigate data efficiency of hybrid loss training, and more precisely how performance changes when varying the number of points at which the data loss and physics loss are evaluated when training the proposed $m\mathrm{GNO} \circ \mathrm{FNO}$ models using a hybrid training loss $\mathcal{L}_{\mathrm{data}} + \lambda \mathcal{L}_{\mathrm{physics}}$. Table 4 shows the results obtained with $m\mathrm{GNO} \circ \mathrm{FNO}$ models on the time-dependent Burgers' equation. We see that performance remains unaffected until the spatial resolution falls below 16 or the temporal resolution falls below 5. This corresponds to $8\times$ spatial subsampling and $5\times$ time subsampling from the original $128 \times 26$ grid, and highlights the data efficiency of the proposed approach: instead of requiring high resolution $128 \times 26$ simulations (3328 points) for the data loss, we can obtain results of similar accuracy by supplementing low resolution simulations (e.g. at $22 \times 9$, i.e. 198 points) with a physics loss evaluated at a higher resolution.

Table 4: Relative L2 data loss and physics loss (computed using Autograd) for the proposed $m\mathrm{GNO} \circ \mathrm{FNO}$ models on Burgers' equation, when training with a hybrid loss. We consider the effect of reducing the resolution at which the data loss is evaluated during training, starting from the original $128 \times 26$ resolution. Each row corresponds to a different spatial resolution (subsampled from 128), while each column corresponds to a different temporal resolution (subsampled from 26). We show in orange the settings where the data loss starts being affected, and in red the settings where the data loss is significantly worse.

| Space / Time | 26 Physics | 26 Data | 13 Physics | 13 Data | 9 Physics | 9 Data | 5 Physics | 5 Data |
|---|---|---|---|---|---|---|---|---|
| 128 | $1.48 \cdot 10^{-4}$ | $3.81 \cdot 10^{-3}$ | $1.85 \cdot 10^{-4}$ | $4.69 \cdot 10^{-3}$ | $1.77 \cdot 10^{-4}$ | $4.80 \cdot 10^{-3}$ | $1.11 \cdot 10^{-4}$ | $6.84 \cdot 10^{-3}$ |
| 64 | $1.73 \cdot 10^{-4}$ | $3.83 \cdot 10^{-3}$ | $1.95 \cdot 10^{-4}$ | $4.89 \cdot 10^{-3}$ | $1.86 \cdot 10^{-4}$ | $4.82 \cdot 10^{-3}$ | $1.17 \cdot 10^{-4}$ | $7.04 \cdot 10^{-3}$ |
| 43 | $1.74 \cdot 10^{-4}$ | $4.04 \cdot 10^{-3}$ | $1.85 \cdot 10^{-4}$ | $4.84 \cdot 10^{-3}$ | $1.88 \cdot 10^{-4}$ | $5.02 \cdot 10^{-3}$ | $1.14 \cdot 10^{-4}$ | $6.86 \cdot 10^{-3}$ |
| 32 | $1.74 \cdot 10^{-4}$ | $4.22 \cdot 10^{-3}$ | $1.85 \cdot 10^{-4}$ | $4.89 \cdot 10^{-3}$ | $1.82 \cdot 10^{-4}$ | $4.91 \cdot 10^{-3}$ | $1.17 \cdot 10^{-4}$ | $6.84 \cdot 10^{-3}$ |
| 26 | $1.82 \cdot 10^{-4}$ | $4.21 \cdot 10^{-3}$ | $1.77 \cdot 10^{-4}$ | $4.82 \cdot 10^{-3}$ | $1.92 \cdot 10^{-4}$ | $4.87 \cdot 10^{-3}$ | $1.13 \cdot 10^{-4}$ | $6.88 \cdot 10^{-3}$ |
| 22 | $1.70 \cdot 10^{-4}$ | $4.03 \cdot 10^{-3}$ | $1.68 \cdot 10^{-4}$ | $4.60 \cdot 10^{-3}$ | $1.72 \cdot 10^{-4}$ | $4.77 \cdot 10^{-3}$ | $1.21 \cdot 10^{-4}$ | $7.03 \cdot 10^{-3}$ |
| 16 | $1.84 \cdot 10^{-4}$ | $8.60 \cdot 10^{-3}$ | $1.99 \cdot 10^{-4}$ | $8.08 \cdot 10^{-3}$ | $1.97 \cdot 10^{-4}$ | $7.27 \cdot 10^{-3}$ | $1.41 \cdot 10^{-4}$ | $8.33 \cdot 10^{-3}$ |
| 11 | $1.86 \cdot 10^{-4}$ | $1.32 \cdot 10^{-2}$ | $1.90 \cdot 10^{-4}$ | $1.17 \cdot 10^{-2}$ | $2.04 \cdot 10^{-4}$ | $1.11 \cdot 10^{-2}$ | $1.40 \cdot 10^{-4}$ | $1.13 \cdot 10^{-2}$ |
| 7 | $1.57 \cdot 10^{-4}$ | $5.67 \cdot 10^{-2}$ | $1.68 \cdot 10^{-4}$ | $5.15 \cdot 10^{-2}$ | $2.01 \cdot 10^{-4}$ | $5.06 \cdot 10^{-2}$ | $1.21 \cdot 10^{-4}$ | $5.12 \cdot 10^{-2}$ |

### 4.1.4 Robustness to Noisy Data

We now investigate the effect of noisiness on reference data when training the proposed $m\mathrm{GNO} \circ \mathrm{FNO}$ models using a hybrid training loss $\mathcal{L}_{\mathrm{data}} + \lambda \mathcal{L}_{\mathrm{physics}}$, both with finite differences and automatic differentiation.

More precisely, instead of the distance to the reference solution $u$, the data loss is based on the distance to a noisy reference solution

$$\tilde{u}(x,t) = u(x,t) + \eta \cdot \varepsilon(x,t) \cdot \int_0^1 \int_0^1 |u(x',t')| \, \mathrm{d}x' \, \mathrm{d}t', \tag{15}$$

where $\eta$ is the noise level, the integral represents the average absolute value of $u(x,t)$, and $\varepsilon(x,t)$ is an independent standard normal random variable for each $(x,t)$, that is $\varepsilon \overset{\mathrm{i.i.d.}}{\sim} \mathcal{N}(0,1)$.

Table 5 illustrates the effect of noise in the reference data on the $m\mathrm{GNO} \circ \mathrm{FNO}$ models trained with a hybrid physics-data loss. The results show that the Autograd approach, our proposed method, maintains very low PDE residuals even as noise increases, demonstrating its robustness and stability. The relative L2 error with respect to the true solution grows with noise level, reflecting the unavoidable discrepancy between noisy training data and the true solution, but increases significantly slower than the training L2 error with respect to the noisy reference. This indicates that the physics loss truly helps the model to learn effectively the dynamics without overfitting to noise.

Compared to finite differences, Autograd achieves slightly lower residuals and errors across noise levels, highlighting its reliability for accurately computing derivatives within the physics loss. These results confirm that the proposed Autograd-based hybrid training is effective and resilient, preserving model accuracy even when the reference data is noisy, while finite differences provide a reasonable but less precise baseline.

Table 5: Effect of injecting noise of different levels $\eta$ (as defined in equation 15) in the reference data when training a $m\mathrm{GNO} \circ \mathrm{FNO}$ on Burgers' equation with a hybrid loss. We report the PDE residual and the relative L2 data loss both to the true reference solution $u$ and to the noisy reference solution $\tilde{u}$. We consider both the cases where the derivatives are computed using Autograd and finite differences in the physics losses.

| Noise Level $\eta$ | PDE Residual | Relative L2 (true) | Training Relative L2 (noisy) |
|---|---|---|---|
| **Autograd** | | | |
| $\eta = 0\%$ (Reference) | $3.43 \cdot 10^{-4}$ | $1.68 \cdot 10^{-3}$ | $1.68 \cdot 10^{-3}$ |
| $\eta = 0.1\%$ | $3.39 \cdot 10^{-4}$ | $1.70 \cdot 10^{-3}$ | $1.90 \cdot 10^{-3}$ |
| $\eta = 0.5\%$ | $3.08 \cdot 10^{-4}$ | $1.99 \cdot 10^{-3}$ | $4.62 \cdot 10^{-3}$ |
| $\eta = 1\%$ | $2.79 \cdot 10^{-4}$ | $2.64 \cdot 10^{-3}$ | $8.74 \cdot 10^{-3}$ |
| $\eta = 2\%$ | $3.01 \cdot 10^{-4}$ | $4.09 \cdot 10^{-3}$ | $1.72 \cdot 10^{-2}$ |
| $\eta = 5\%$ | $4.93 \cdot 10^{-4}$ | $8.41 \cdot 10^{-3}$ | $4.29 \cdot 10^{-2}$ |
| $\eta = 10\%$ | $5.73 \cdot 10^{-4}$ | $1.31 \cdot 10^{-2}$ | $8.58 \cdot 10^{-2}$ |
| $\eta = 20\%$ | $9.79 \cdot 10^{-4}$ | $2.22 \cdot 10^{-2}$ | $1.70 \cdot 10^{-1}$ |
| $\eta = 50\%$ | $1.67 \cdot 10^{-3}$ | $4.39 \cdot 10^{-2}$ | $3.97 \cdot 10^{-1}$ |
| **Finite Differences** | | | |
| $\eta = 0\%$ (Reference) | $4.78 \cdot 10^{-4}$ | $1.35 \cdot 10^{-3}$ | $1.35 \cdot 10^{-3}$ |
| $\eta = 0.1\%$ | $4.88 \cdot 10^{-4}$ | $1.40 \cdot 10^{-3}$ | $1.63 \cdot 10^{-3}$ |
| $\eta = 0.5\%$ | $5.00 \cdot 10^{-4}$ | $2.24 \cdot 10^{-3}$ | $4.73 \cdot 10^{-3}$ |
| $\eta = 1\%$ | $5.30 \cdot 10^{-4}$ | $3.54 \cdot 10^{-3}$ | $9.10 \cdot 10^{-3}$ |
| $\eta = 2\%$ | $6.28 \cdot 10^{-4}$ | $5.81 \cdot 10^{-3}$ | $1.78 \cdot 10^{-2}$ |
| $\eta = 5\%$ | $8.23 \cdot 10^{-4}$ | $1.12 \cdot 10^{-2}$ | $4.40 \cdot 10^{-2}$ |
| $\eta = 10\%$ | $1.16 \cdot 10^{-3}$ | $1.85 \cdot 10^{-2}$ | $8.74 \cdot 10^{-2}$ |
| $\eta = 20\%$ | $1.71 \cdot 10^{-3}$ | $3.07 \cdot 10^{-2}$ | $1.72 \cdot 10^{-1}$ |
| $\eta = 50\%$ | $2.98 \cdot 10^{-3}$ | $6.08 \cdot 10^{-2}$ | $4.01 \cdot 10^{-1}$ |

### 4.2 Nonlinear Poisson Equation

### 4.2.1 Problem Description

The Poisson equation is a fundamental PDE that appears in numerous applications in science due to its ability to model phenomena with spatially varying and nonlinear behaviors. We consider the nonlinear Poisson equation with varying source terms, boundary conditions, and geometric domain,

$$\nabla \cdot \left[(1 + 0.1u(\mathbf{x})^2)\nabla u(\mathbf{x})\right] = f(\mathbf{x}) \qquad\qquad \mathbf{x} \in \Omega \qquad (16)$$

$$u(\mathbf{x}) = b(\mathbf{x}) \qquad\qquad \mathbf{x} \in \partial\Omega \qquad (17)$$

where $u \in \mathbb{R}$ and $\Omega \subset \mathbb{R}^2$. The domain $\Omega$ is centered at the origin and defined in polar coordinates with varying radius about the origin

$$r(\theta) = r_0[1 + c_1 \cos(4\theta) + c_2 \cos(8\theta)],$$

where the parameters $c_1$ and $c_2$ are drawn from a uniform distribution on $(-0.2, 0.2)$. The source term $f$ is a sum of radial basis functions $f(\mathbf{x}) = \sum_{i=1}^{3} \beta_i \exp \|\mathbf{x} - \mu_i\|_2^2$, where $\beta_i \in \mathbb{R}$ and $\mu_i \in \mathbb{R}^2$ are both drawn from standard normal distributions. The boundary condition $b$ is a periodic function, defined in polar coordinates as $b_0 + \frac{1}{4}[b_1 \cos(\theta) + b_2 \sin(\theta) + b_3 \cos(2\theta) + b_4 \sin(2\theta)]$, where the parameters $b_i$ are drawn from a uniform distribution on $(-1, 1)$. This is the setting used by Qin et al. (2022).

The mesh coordinates, signed distance functions, source terms, and boundary conditions, are passed through a GINO model of the form

$$\mathcal{G}_{mGNO}^{decoder} \circ \mathcal{G}_{FNO} \circ \mathcal{G}_{GNO}^{encoder} \qquad (18)$$

to produce an approximation to the solution $u$. We minimize the PDE residual and boundary condition loss,

$$\mathcal{L}_{\text{Poisson}}(v) \;=\; \left\|\nabla \cdot \left[(1 + 0.1v^2)\nabla v\right] - f\right\|_{L^2(\Omega)}^2 \;+\; \alpha \left\|v - b\right\|_{L^2(\partial\Omega)}^2. \qquad (19)$$

Examples of predictions made by the trained $m$GINO model for a variety of geometries are displayed in Figure 6, below the corresponding reference solutions.

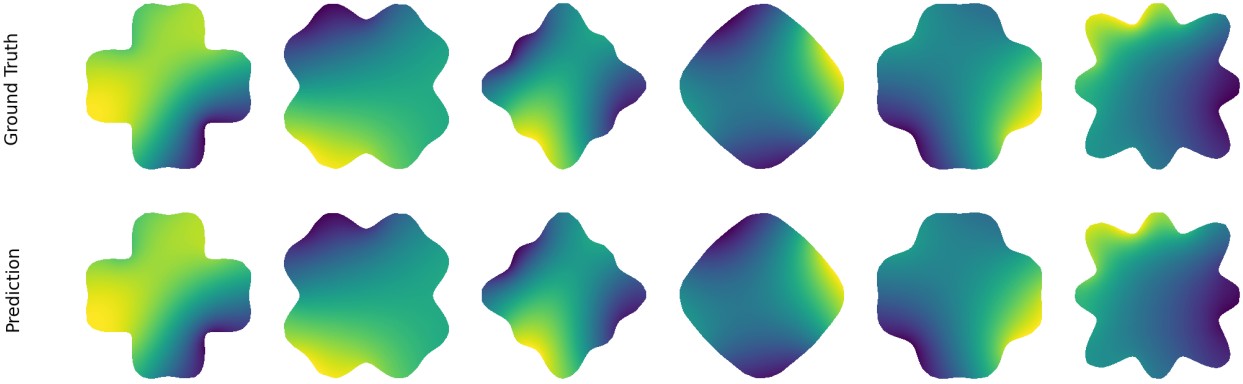

Figure 6: Comparison of reference solutions to the nonlinear Poisson equation on various geometries *(top)* with the corresponding $m$GINO predictions *(bottom)*.

### 4.2.2 Comparison to Baselines

The results in Figure 7 show the trade-off between inference time and accuracy. $m$GINO achieves a relative squared error 2-3 orders of magnitude lower than Meta-PDE (i.e. initialized PINNs) for a comparable running time, and a speedup of 20-25× compared to the solver for similar relative accuracy. In addition, $m$GINO is more consistent across different instances, with smaller variations in errors compared to Meta-PDE.

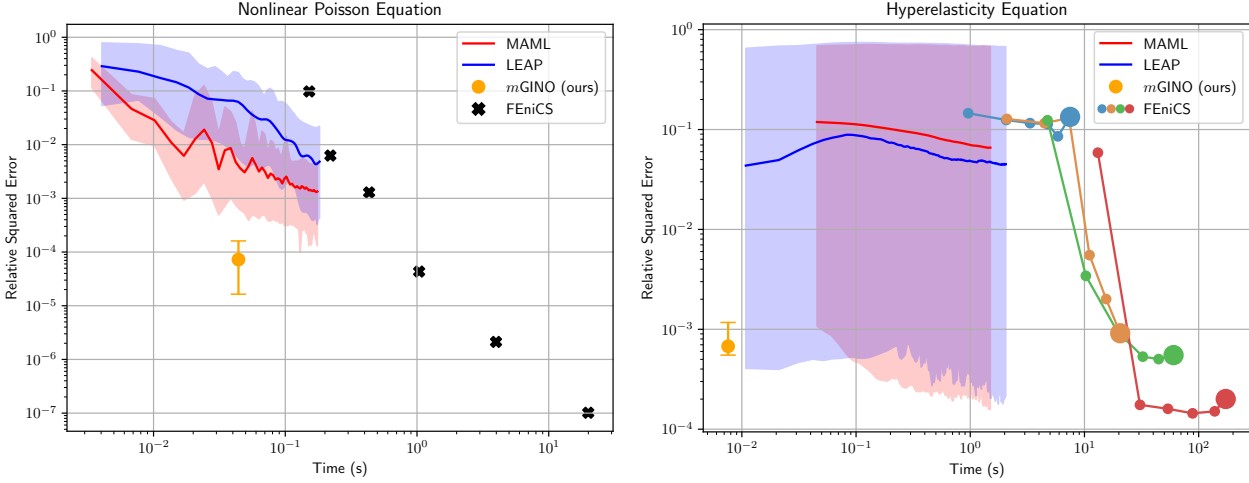

Figure 7: Computational time for inference versus accuracy for the nonlinear Poisson and hyperelasticity equations. We compare the proposed approach with Meta-PDE (MAML and LEAP) and with the baseline FEniCS solver. The FEniCS solver was used with 6 different resolutions for the Poisson equation, and 4 different resolutions (iteratively tries to refine the solution) for the hyperelasticity equation. The mean values across 200 instances are plotted, with the shaded regions representing the min and max error values.

### 4.2.3 Using Different Training Losses

We trained models using a data loss, a physics loss, and a hybrid loss (data and physics). Table 6 shows that using a physics loss has a comparable data error to the data-driven approach, while achieving a PDE residual 4-5 orders of magnitude lower. Fine-tuning the trained data-driven model using $\mathcal{L}_{\mathrm{Poisson}}$ did not work well, as its physics loss is 7 orders of magnitude higher than its data loss, and attempts at lowering the physics loss only worked by completely sacrificing the data loss.

Table 6: Data and PDE losses of $m$GINO trained with different losses $\mathcal{L}$ for the Poisson equation.

|  | $\mathcal{L}_{\mathrm{data}}$ | $\mathcal{L}_{\mathrm{data}} + \lambda\mathcal{L}_{\mathrm{physics}}$ | $\mathcal{L}_{\mathrm{physics}}$ |
|---|---|---|---|
| MSE Data Loss | $1.36 \cdot 10^{-5}$ | $2.29 \cdot 10^{-5}$ | $1.39 \cdot 10^{-5}$ |
| PDE Residual Loss | $3.55 \cdot 10^{2}$ | $2.07 \cdot 10^{-2}$ | $7.21 \cdot 10^{-3}$ |

When training only with data loss, the predicted solutions have discontinuities at higher resolutions coinciding with circles of radius $r$ centered at the latent query points of the GINO (see Figure 8(a)). These discontinuities lead to high derivatives and inaccurate physics losses, regardless of the differentiation method used (see Figures 8(b)(c)). This happens despite the low data MSE, indicating that only training the $m$GINO model with data loss is not sufficient to capture the solution correctly at higher resolutions.

Recall that the GNO's kernel integration can be viewed as an aggregation of messages if we construct a graph on the spatial domain of the PDE, as described in Li et al. (2020b). The *mean* aggregation is given by

$$v_{t+1} = \sigma\left(Wv_t(x) + \frac{1}{|N(x)|}\sum_{y \in N(x)}\kappa_\phi(e(x,y))v_t(y)\right)$$

where $v_t(x) \in \mathbb{R}^n$ are the node features, $e(x,y) \in \mathbb{R}^{n_e}$ are the edge features, $W \in \mathbb{R}^{n \times n}$ is learnable, $N(x)$ is the neighborhood of $x$, and $\kappa_\phi(e(x,y))$ is a neural network mapping edge features to a matrix in $R^{n \times n}$. When using a differentiable weight function, points in $N(x)$ at the edge of the neighborhood have near-zero weights but still contribute to the denominator $|N(x)|$. Thus, as the query point $x$ moves slightly, additional neighbors get included or excluded with near-zero weights, thereby introducing the discontinuities we see in Figure 8(a). Using a *sum* aggregation for the output GNO's kernel integration mitigates the rigid patterns. When training with a physics loss, the patterns disappear, as shown in Figure 9, while the MSE for the predictions remains of the same order of magnitude.

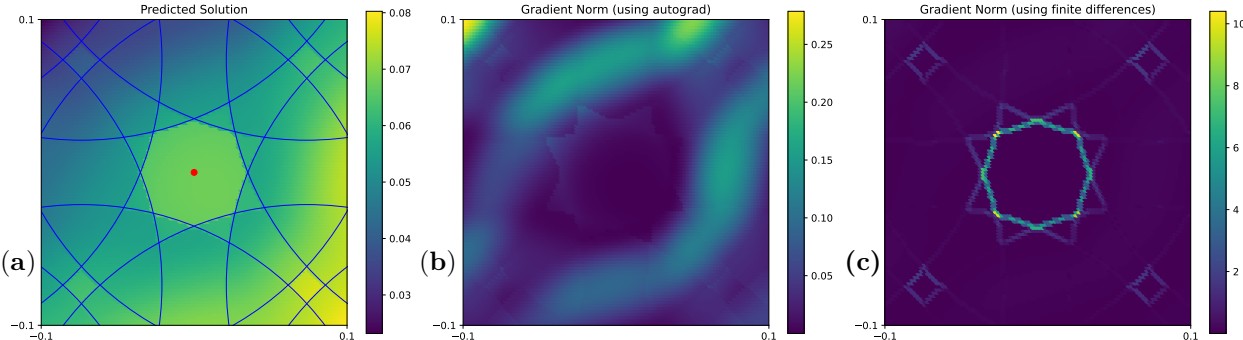

Figure 8: **(a)** Prediction of a $m$GINO **trained using data loss only** for the nonlinear Poisson equation. This is the predicted solution **evaluated at a high resolution** on a small patch centered at a latent query point of the $m$GNO. The prediction exhibits discontinuities that coincide with the circles of radius $r$ (blue lines) centered at the neighboring latent query points. **(b)(c)** Norm of the gradient of the predicted solution shown in (a), computed using automatic differentiation (in (b)) and finite differences (in (c)).

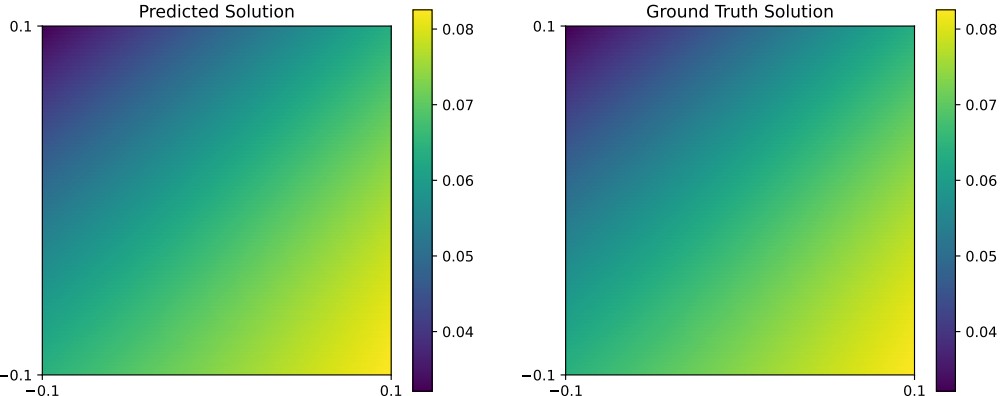

Figure 9: Comparison between the prediction of a $m$GINO model *(left)* **trained using a hybrid loss**, $\mathcal{L} = \mathcal{L}_{\text{data}} + \lambda \mathcal{L}_{\text{physics}}$ and the corresponding ground truth solution *(right)* for the nonlinear Poisson equation. The prediction and ground truth solutions are **evaluated at a high resolution** on a small patch centered at a latent query point of the $m$GNO. Using the physics loss regularized the higher resolution prediction, and removed the visible discontinuity patterns shown in Figure 8(a).

#### 4.2.4 Nonlinear Poisson PDE Residual Loss with Finite Differences

Figure 10 shows gradients norms on a domain patch, computed using Autograd and finite differences at $16 \times 16$ and higher resolutions (both displayed at $16 \times 16$). The finite difference gradient norms at lower resolution in (c) differ completely from those obtained using Autograd (b)(e), and high-resolution finite differences (f), while the latter are very similar. In addition, inaccuracies can be further amplified with finite differences on unstructured point clouds in regions with a low density of points. The Autograd $m$GINOs were trained successfully with a density of randomly located points on the whole domain slightly lower than the density here with $16^2$ points on the domain patch displayed in Figure 10. Given that numerical errors made on derivatives are amplified in physics losses, $\mathcal{L}_{\text{Poisson}}$ does not provide enough information to move towards physically plausible solutions when computed using finite differences on unstructured point clouds at the same resolution at which Autograd $m$GINOs models were trained successfully. This showcases how Autograd $m$GINO can be used to obtain surrogate models using physics losses at resolutions at which finite differences are not accurate enough. We estimate that at least $9\times$ more points would be necessary to compute reasonable finite difference derivatives. Higher resolution finite difference computations typically require a higher computational time and memory requirement (see Appendix F for a comparison of training times at different resolutions).

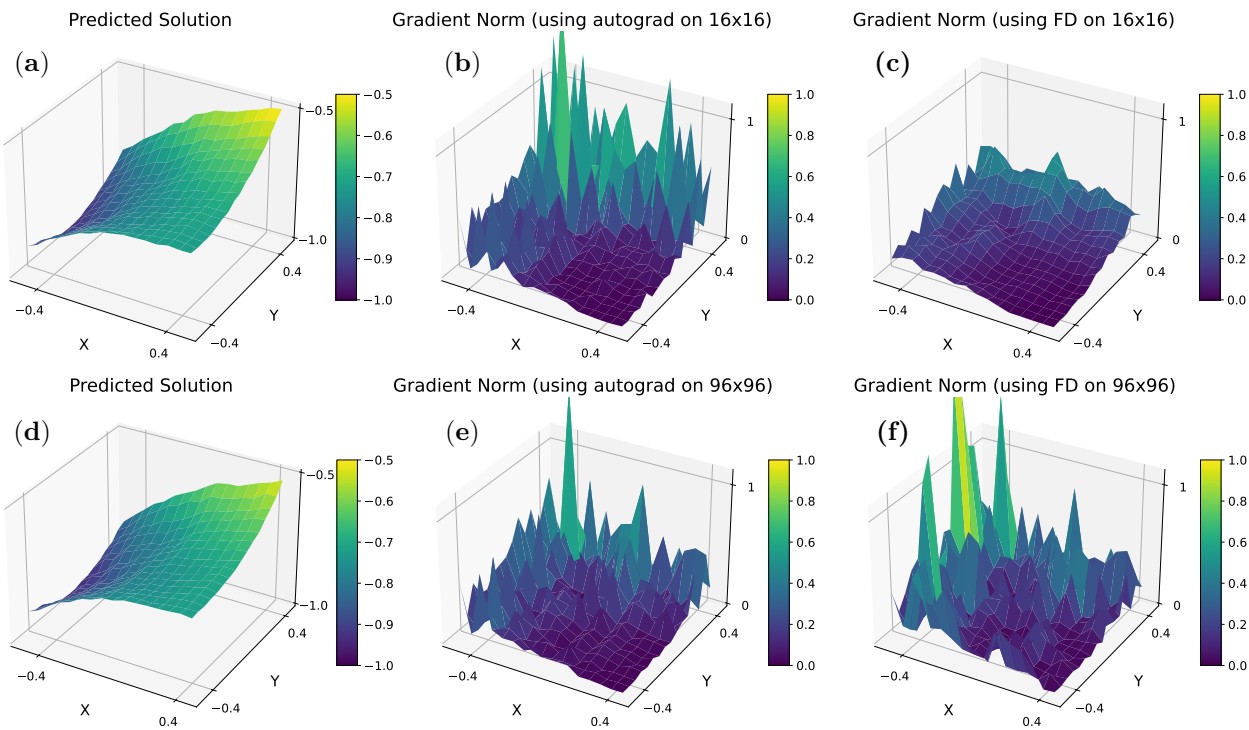

Figure 10: Prediction of a $m$GINO model and the norm of its gradient, computed on a $16 \times 16$ regular grid *(top row)* and a $96 \times 96$ regular grid *(bottom row)* using Autograd and finite differences (FD). All plots are displayed on a $16 \times 16$ grid for ease of qualitative comparison.

### 4.2.5 Data Efficiency

We now investigate the data efficiency of hybrid loss training, and more precisely how performance changes when varying the number of points at which the data loss and physics loss are evaluated when using a hybrid training loss $\mathcal{L}_{\text{data}} + \lambda \mathcal{L}_{\text{physics}}$ for the nonlinear Poisson equation. Table 7 displays the results obtained with the proposed $m$GINO model, showing that both the data and physics losses worsen when the number of points used for the physics loss during training decreases. However, performance is not very heavily impacted when the number of points used for the data loss during training is reduced.

Table 7: Data loss (MSE) and physics loss (computed using Autograd) for the proposed $m$GINO models on the Poisson equation, when training with a hybrid loss. We vary the resolutions at which the data loss and physics losses are evaluated during training. Each row corresponds to a different number of points used for the physics loss, while each column corresponds to a different number of points used for the data loss.

| Physics \ Data | 400 points | 200 points | 50 points |
|---|---|---|---|
|  | Physics \| Data | Physics \| Data | Physics \| Data |
| 800 points | $2.63 \cdot 10^{-2} \mid 8.28 \cdot 10^{-5}$ | $3.31 \cdot 10^{-2} \mid 3.89 \cdot 10^{-5}$ | $3.82 \cdot 10^{-2} \mid 6.73 \cdot 10^{-5}$ |
| 400 points | $2.13 \cdot 10^{-1} \mid 1.10 \cdot 10^{-4}$ | $1.22 \cdot 10^{-1} \mid 9.20 \cdot 10^{-5}$ | $1.39 \cdot 10^{-1} \mid 1.29 \cdot 10^{-4}$ |
| 200 points | $3.75 \cdot 10^{-1} \mid 2.40 \cdot 10^{-4}$ | $2.55 \cdot 10^{-1} \mid 2.92 \cdot 10^{-4}$ | $3.33 \cdot 10^{-1} \mid 5.18 \cdot 10^{-4}$ |

### 4.2.6 Ablation Study on GNO Radius and Weight Function

We investigate how performance changes as the $m$GNO radius changes for $m$GINO models, and also compare the performance of the different weight functions (9)-(12) on the nonlinear Poisson equation. Note that the radius cutoff used in the weight functions of $m$GNOs is the same as the $m$GNO radius to avoid the additional cost of carrying a second neighbor search. The $m$GNO radius $r$ is an important hyperparameter to tune. It needs to be large enough so that the ball $B_r(x)$ contains sufficiently many other latent query points, and a value too large can lead to prohibitive computational and memory costs. We denote the number of latent query points in $B_r(x)$ by $\#|B_r(x)|$. For the nonlinear Poisson equation, we are using a regular 2D latent space grid, so $B_r(x)$ will only contain a single latent query point ($x$ itself) until $r$ is large enough for $B_r(x)$ to contain one extra latent query point in each direction. $B_r(x)$ will progressively contain more latent query points by thresholds as the radius increases further, together with computational time and memory cost.

Table 8 shows the results obtained for $m$GINO models with different weight functions for values of $r$ such that $\#|B_r(x)| = 3^2, 5^2, 7^2, 9^2, 15^2$. As expected, performance is poor when the radius is too small. On the other hand, when $r$ becomes too large, the computational and memory costs significantly increase while performance deteriorates. For this numerical experiment, the best performance was achieved with a radius for which $\#|B_r(x)| = 7^2$, and $w_{\text{half\_cos}}$ and $w_{\text{quartic}}$ led to the best performance.

Table 8: MSE of $m$GINO models trained on only PDE loss, with different GNO radii and weight functions.

|  |  | bump | half_cos | quartic | octic |
|---|---|---|---|---|---|
| $\#|B_r(x)| = 9$ | $(r = 0.0875)$ | $3.64 \cdot 10^{-1}$ | $3.41 \cdot 10^{-1}$ | $2.20 \cdot 10^{-2}$ | $3.48 \cdot 10^{-1}$ |
| $\#|B_r(x)| = 25$ | $(r = 0.13125)$ | $4.51 \cdot 10^{-5}$ | $3.59 \cdot 10^{-5}$ | $3.95 \cdot 10^{-5}$ | $4.24 \cdot 10^{-5}$ |
| $\#|B_r(x)| = 49$ | $(r = 0.175)$ | $1.40 \cdot 10^{-4}$ | $\mathbf{1.39 \cdot 10^{-5}}$ | $2.93 \cdot 10^{-5}$ | $9.49 \cdot 10^{-5}$ |
| $\#|B_r(x)| = 81$ | $(r = 0.21875)$ | $5.35 \cdot 10^{-4}$ | $1.01 \cdot 10^{-4}$ | $7.79 \cdot 10^{-5}$ | $1.76 \cdot 10^{-4}$ |
| $\#|B_r(x)| = 225$ | $(r = 0.35)$ | $3.92 \cdot 10^{-3}$ | $9.36 \cdot 10^{-4}$ | $4.28 \cdot 10^{-3}$ | $6.59 \cdot 10^{-4}$ |

## 4.3 Hyperelasticity Equation

### 4.3.1 Problem Description

The hyperelasticity equation models the shape deformation under external forces of hyperelastic materials (e.g. rubber) for which the stress-strain relation is highly nonlinear. We consider the deformation of a homogeneous and isotropic hyperelastic material when compressed uniaxially (assuming no body and traction forces), and learn the final deformation displacement $u$ mapping the initial reference position to the deformed location. More precisely, we consider the deformation of a two-dimensional porous hyperelastic material under compression, as in Overvelde & Bertoldi (2014) and Qin et al. (2022). We keep the pores circular and fix the distance between the pore centers, so that the size of the pore is the only varied parameter. The size of the pores determines the porosity of the structure and affects the macroscopic deformation behavior of the structure. Examples of final deformation displacement fields are shown in Figure 11.

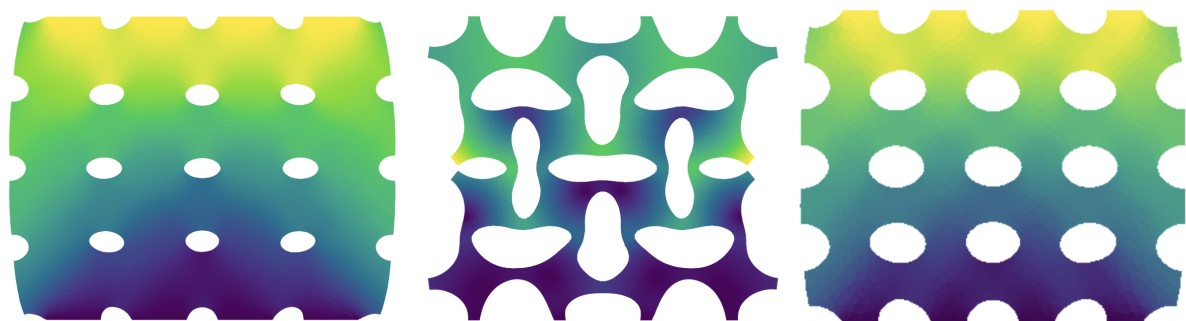

Figure 11: Final deformation displacement for several instances of the hyperelasticity dataset.

The mesh coordinates and signed distance functions are passed through a GINO model of the form

$$\mathcal{G}_{m\text{GNO}}^{decoder} \circ \mathcal{G}_{\text{FNO}} \circ \mathcal{G}_{\text{GNO}}^{encoder} \tag{20}$$

to produce an approximation to the solution $u$ to the hyperelasticity equations.

The solution can be obtained as the minimizer of the total Helmholtz free energy of the system $\int_{\Omega} \psi \, d\mathbf{x}$. Here $\psi$ denotes the Helmholtz free energy relating the Piola–Kirchhoff stress $P$ with the deformation gradient $F$ via $P = \frac{d\psi}{dF}$. Instead of minimizing the PDE loss from the strong form of the hyperelasticity equation, we minimize the Helmholtz free energy of the system by randomly sampling collocation points from the PDE domain and the Dirichlet boundary and using these points to form a Monte Carlo estimate of the total Helmholtz free energy. In addition, we add a weighted boundary loss term.

### 4.3.2 Numerical Results

We trained models using a data loss, a physics loss, and a hybrid loss. Table 9 shows that using a physics loss is critical for this problem, as the data-driven approach achieves a low data loss, but the predictions do not satisfy the physics at all. In contrast, both the physics-only and hybrid approaches achieve low data and PDE errors. As for the Poisson equation, fine-tuning the trained data-driven model using the physics loss did not work, due to the high physics loss of the data-driven model. As in Section 4.2.4, Autograd enables computation of accurate derivatives at resolutions where finite differences are not accurate.

Table 9: Data and PDE losses of $m$GINO trained with different losses $\mathcal{L}$ for the hyperelasticity equation.

|  | $\mathcal{L}_{\text{data}}$ | $\mathcal{L}_{\text{data}} + \lambda \mathcal{L}_{\text{physics}}$ | $\mathcal{L}_{\text{physics}}$ |
|---|---|---|---|
| MSE Data Loss | $3.35 \cdot 10^{-7}$ | $9.69 \cdot 10^{-7}$ | $9.42 \cdot 10^{-5}$ |
| PDE Residual Loss | $2.97 \cdot 10^{26}$ | $1.57 \cdot 10^{-2}$ | $1.21 \cdot 10^{-2}$ |

A comparison to baselines is displayed in Figure 7. $m$GINO achieves a relative squared error 2 orders of magnitude lower than Meta-PDE (i.e. initialized PINNs) with a slightly faster running time, and achieves a speedup of 3000-4000$\times$ compared to the FEniCS solver for similar accuracy. In addition, $m$GINO achieves consistent results across different samples (all within a single order of magnitude) while Meta-PDE results span more than 3 orders of magnitude. This high variation in Meta-PDE results is likely caused by the difficulty disparity across samples, as samples with larger pores are harder to resolve.

### 4.4 Navier–Stokes Equations

### 4.4.1 Problem Setting

Finally, we consider the lid cavity flow governed by the Navier–Stokes equations

$$\partial_t u(x,t) + u(x,t) \cdot \nabla u(x,t) = -\frac{1}{\rho} \nabla p(x,t) + \frac{1}{Re} \Delta u(x,t), \qquad x \in (0,1)^2, t \in (0,T]$$
$$\nabla \cdot u(x,t) = 0, \qquad x \in (0,1)^2, t \in [0,T] \tag{21}$$
$$u(x,0) = u_0(x), \qquad x \in (0,1)^2$$

We study the standard cavity flow on domain $D = (0,1)^2$ with $T = 10$ seconds, where $u$ is velocity, $p$ is pressure, $\rho = 1$ is density, and $Re = 500$. We assume zero initial conditions, and the no-slip boundary condition where $u(x,t) = (0,0)$ at the left, bottom, and right walls and $u(x,t) = (1,0)$ on top. We also start from zero initial conditions, $u(0,t) = 0$ for $x \in (0,1)^2$. We are interested in learning the solution for the time interval $[5,10]$. Figure 12 shows the solution to this equation at a few timesteps. The main challenge lies in handling the boundary conditions within the velocity–pressure formulation.

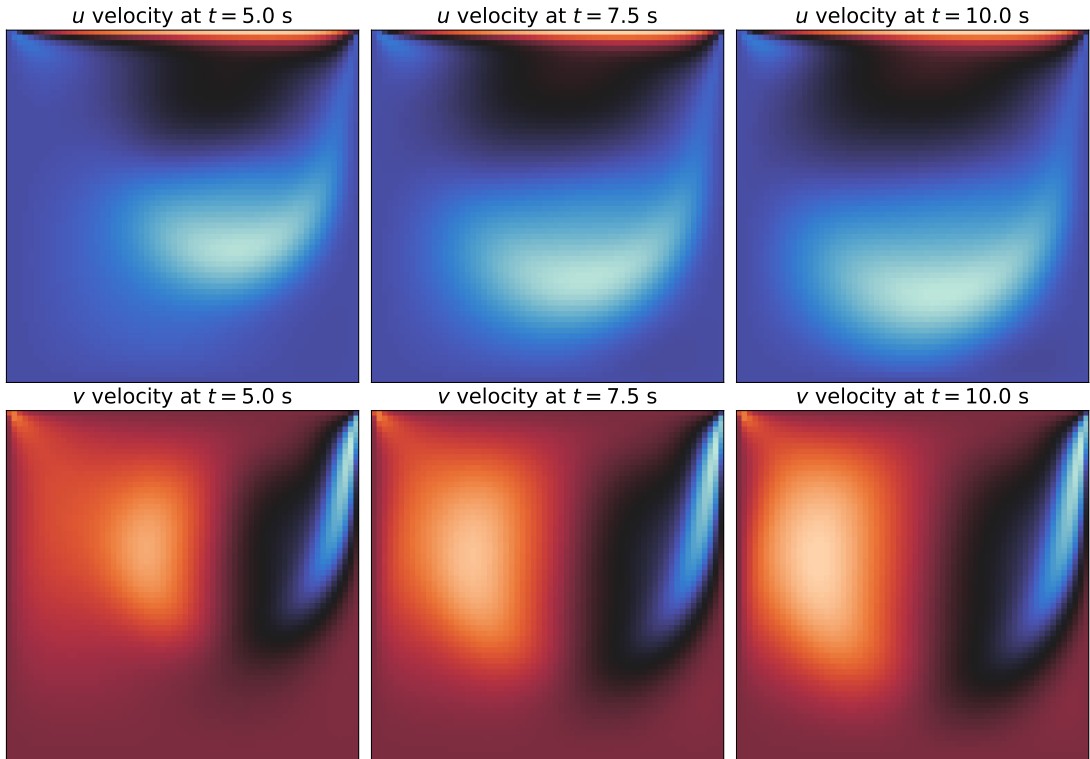

Figure 12: Example of solution for the lid cavity flow governed by the Navier–Stokes equations.

### 4.4.2 Results

We first trained $m$GNO $\circ$ FNO models using a data-only loss, a physics-only loss, and a hybrid loss. To facilitate the tuning of the loss coefficients for this numerical experiment, we employ ReLoBRaLo (Relative Loss Balancing with Random Lookback) (Bischof & Kraus, 2021) to adaptively update and balance the various loss coefficients in the total loss. ReLoBRaLo uses the history of loss decay across random lookback intervals to achieve uniform progress across the different terms and faster convergence overall. We also investigate how performance changes when varying the number of points at which the data and physics losses are evaluated during training when using a hybrid training loss $\mathcal{L}_{\text{data}} + \lambda \mathcal{L}_{\text{physics}}$ for the Navier–Stokes equations. Table 10 displays the results obtained.

We see that access to data is essential, as training solely on the PDE residual leads to high data loss. Also, training with data on a fixed grid results in poor generalization to higher resolutions. By contrast, with Autograd, a small number of randomly sampled points is sufficient to get excellent results. This demonstrates that random sampling allows the Autograd $m$GNO $\circ$ FNO to capture correct underlying physics instead of overfitting to a fixed grid. Also, including a physics loss is critical. Models trained only on data loss exhibit extremely large PDE residuals, while incorporating the physics term, even at low resolution with randomly sampled points, ensures good PDE fidelity. This indicates that the physics loss guides the network toward physically consistent solutions, and very high-resolution PDE evaluation is not always necessary.

Finally, none of the results obtained with finite differences were satisfactory. It is worth noting that these were trained with a physics loss on a fixed grid (given the added complexity and reduced reliability of non-uniform finite differences), which can limit generalization across resolutions. However, even accounting for this, using finite differences physics losses led to results significantly worse than training purely on data, both in terms of PDE residual and data loss at the test resolutions. This reveals a clear mismatch between the finite-difference PDE loss and the reference data, suggesting that the finite-difference derivatives were not sufficiently accurate at the training resolutions, even when using as many as 50,000 randomly sampled points.

Table 10: Results of training $m$GNO $\circ$ FNO models with a hybrid loss for the Navier–Stokes equations, using varying resolutions for the physics term (PDE Res.) and the data term (Data Res.) during training. When the resolution is denoted by a single number $N$, it indicates that $N$ points are randomly sampled at each training iteration. In contrast, when the resolution is written as $N_x \times N_y \times N_t$, it refers to using a fixed grid with the specified spatial and temporal resolutions. The table reports both PDE residuals and data losses at the training resolutions, and at the testing resolutions (fixed to $32 \times 32 \times 16$ for the PDE residual and $64 \times 64 \times 50$ for the data loss). Derivatives are computed with Autograd and finite differences. Cells highlighted indicate suboptimal results (orange) or poor generalization (red) at the test resolution.

### AUTOGRAD

| PDE Res. | Data Res. | PDE (Train) | PDE (Test) | Data (Train) | Data (Test) |
|---|---|---|---|---|---|
| 10000 | $32 \times 32 \times 50$ | $3.25 \cdot 10^{-5}$ | $6.99 \cdot 10^{-5}$ | $2.91 \cdot 10^{-4}$ | $1.50 \cdot 10^{-2}$ |
| 10000 | 50000 | $1.71 \cdot 10^{-6}$ | $2.13 \cdot 10^{-6}$ | $1.06 \cdot 10^{-4}$ | $1.05 \cdot 10^{-4}$ |
| 10000 | 10000 | $1.52 \cdot 10^{-6}$ | $2.04 \cdot 10^{-6}$ | $1.11 \cdot 10^{-4}$ | $1.09 \cdot 10^{-4}$ |
| 10000 | 1000 | $2.51 \cdot 10^{-5}$ | $3.11 \cdot 10^{-5}$ | $2.53 \cdot 10^{-4}$ | $2.27 \cdot 10^{-4}$ |
| 10000 | 100 | $1.70 \cdot 10^{-4}$ | $1.80 \cdot 10^{-4}$ | $1.32 \cdot 10^{-3}$ | $1.87 \cdot 10^{-3}$ |
| 10000 | —— | $1.16 \cdot 10^{-7}$ | $2.82 \cdot 10^{-7}$ | —— | $2.04 \cdot 10^{-2}$ |
| 1000 | $32 \times 32 \times 50$ | $2.74 \cdot 10^{-7}$ | $6.28 \cdot 10^{-7}$ | $1.57 \cdot 10^{-5}$ | $1.50 \cdot 10^{-2}$ |
| 1000 | 50000 | $2.48 \cdot 10^{-7}$ | $5.82 \cdot 10^{-7}$ | $5.51 \cdot 10^{-5}$ | $5.79 \cdot 10^{-5}$ |
| 1000 | 10000 | $3.59 \cdot 10^{-7}$ | $6.82 \cdot 10^{-7}$ | $5.44 \cdot 10^{-5}$ | $5.28 \cdot 10^{-5}$ |
| 1000 | 1000 | $6.74 \cdot 10^{-6}$ | $8.02 \cdot 10^{-6}$ | $1.48 \cdot 10^{-4}$ | $1.31 \cdot 10^{-4}$ |
| 1000 | 100 | $3.35 \cdot 10^{-4}$ | $4.13 \cdot 10^{-4}$ | $3.55 \cdot 10^{-3}$ | $3.95 \cdot 10^{-3}$ |
| 1000 | —— | $4.40 \cdot 10^{-6}$ | $1.02 \cdot 10^{-5}$ | —— | $2.84 \cdot 10^{-2}$ |
| 100 | $32 \times 32 \times 50$ | $1.03 \cdot 10^{-7}$ | $2.38 \cdot 10^{-7}$ | $6.58 \cdot 10^{-6}$ | $1.57 \cdot 10^{-2}$ |
| 100 | 50000 | $3.85 \cdot 10^{-6}$ | $5.50 \cdot 10^{-6}$ | $3.49 \cdot 10^{-5}$ | $3.53 \cdot 10^{-5}$ |
| 100 | 10000 | $1.98 \cdot 10^{-6}$ | $2.47 \cdot 10^{-6}$ | $8.01 \cdot 10^{-5}$ | $8.06 \cdot 10^{-5}$ |
| 100 | 1000 | $1.97 \cdot 10^{-6}$ | $2.74 \cdot 10^{-6}$ | $8.59 \cdot 10^{-5}$ | $7.69 \cdot 10^{-5}$ |
| 100 | 100 | $6.64 \cdot 10^{-6}$ | $8.67 \cdot 10^{-6}$ | $2.50 \cdot 10^{-4}$ | $2.32 \cdot 10^{-4}$ |
| 100 | —— | $2.92 \cdot 10^{-7}$ | $3.39 \cdot 10^{-7}$ | —— | $2.01 \cdot 10^{-2}$ |
| 25 | $32 \times 32 \times 50$ | $9.28 \cdot 10^{-8}$ | $1.83 \cdot 10^{-7}$ | $8.04 \cdot 10^{-6}$ | $1.50 \cdot 10^{-2}$ |
| 25 | 50000 | $9.98 \cdot 10^{-7}$ | $9.55 \cdot 10^{-7}$ | $6.63 \cdot 10^{-5}$ | $6.84 \cdot 10^{-5}$ |
| 25 | 10000 | $2.59 \cdot 10^{-7}$ | $1.16 \cdot 10^{-6}$ | $1.07 \cdot 10^{-4}$ | $1.05 \cdot 10^{-4}$ |
| 25 | 1000 | $1.27 \cdot 10^{-6}$ | $1.63 \cdot 10^{-6}$ | $1.16 \cdot 10^{-4}$ | $1.35 \cdot 10^{-4}$ |
| 25 | 100 | $2.46 \cdot 10^{-5}$ | $3.53 \cdot 10^{-5}$ | $2.04 \cdot 10^{-4}$ | $3.08 \cdot 10^{-4}$ |
| 25 | —— | $1.96 \cdot 10^{-7}$ | $1.02 \cdot 10^{-6}$ | —— | $1.92 \cdot 10^{-2}$ |

### FINITE DIFFERENCES

| PDE Res. | Data Res. | PDE (Train) | PDE (Test) | Data (Train) | Data (Test) |
|---|---|---|---|---|---|
| $32 \times 32 \times 25$ | $32 \times 32 \times 50$ | $9.14 \times 10^{-4}$ | $1.09 \times 10^{6}$ | $3.07 \times 10^{-3}$ | $4.29 \times 10^{-2}$ |
| $16 \times 16 \times 25$ | $32 \times 32 \times 50$ | $5.16 \times 10^{-4}$ | $3.42 \times 10^{5}$ | $1.78 \times 10^{-3}$ | $4.28 \times 10^{-2}$ |
| $16 \times 16 \times 12$ | $32 \times 32 \times 50$ | $1.76 \times 10^{-4}$ | $2.45 \times 10^{7}$ | $8.28 \times 10^{-4}$ | $4.12 \times 10^{-2}$ |
| $32 \times 32 \times 25$ | 50000 | $3.58 \times 10^{-3}$ | $1.93 \times 10^{5}$ | $3.43 \times 10^{-3}$ | $3.51 \times 10^{-3}$ |
| $32 \times 32 \times 25$ | 10000 | $3.20 \times 10^{-3}$ | $8.95 \times 10^{4}$ | $3.73 \times 10^{-3}$ | $3.54 \times 10^{-3}$ |

### NO PHYSICS LOSS

| PDE Res. | Data Res. | PDE (Train) | PDE (Test) | Data (Train) | Data (Test) |
|---|---|---|---|---|---|
| —— | $32 \times 32 \times 50$ | —— | $3.55 \cdot 10^{5}$ | $5.30 \cdot 10^{-7}$ | $1.56 \cdot 10^{-2}$ |
| —— | 50000 | —— | $2.26 \cdot 10^{4}$ | $1.87 \cdot 10^{-6}$ | $1.82 \cdot 10^{-6}$ |
| —— | 10000 | —— | $2.83 \cdot 10^{4}$ | $1.97 \cdot 10^{-6}$ | $1.91 \cdot 10^{-6}$ |
| —— | 1000 | —— | $2.47 \cdot 10^{5}$ | $3.84 \cdot 10^{-6}$ | $3.51 \cdot 10^{-6}$ |
| —— | 100 | —— | $2.83 \cdot 10^{4}$ | $4.18 \cdot 10^{-6}$ | $6.54 \cdot 10^{-6}$ |

### 4.5 Airfoil Inverse Design

#### 4.5.1 Description of the Forward Mapping

We consider the transonic flow over an airfoil (ignoring the viscous effect), governed by the Euler equation,

$$\frac{\partial \rho^f}{\partial t} + \nabla \cdot (\rho^f \mathbf{v}) = 0, \tag{22}$$

$$\frac{\partial E}{\partial t} + \nabla \cdot ((E + p)\mathbf{v}) = 0, \tag{23}$$

$$\frac{\partial \rho^f \mathbf{v}}{\partial t} + \nabla \cdot (\rho^f \mathbf{v} \otimes \mathbf{v} + p\mathbb{I}) = 0, \tag{24}$$

where $\rho^f$ is the fluid density, $\mathbf{v}$ is the velocity vector, $p$ is the pressure, and $E$ is the total energy. The far-field boundary condition is $\rho_\infty = p_\infty = 1$, $M_\infty = 0.8$, $AoA = 0$, where $M_\infty$ is the Mach number and $AoA$ is the angle of attack, and the no-penetration condition is imposed at the airfoil.

We use the same dataset as Li et al. (2022), where the shape parameterization of the airfoil follows the design element approach (Farin, 2014). The initial NACA-0012 shape is mapped onto a 'cubic' design element with 8 control nodes in the vertical direction with prior $d \sim \mathbb{U}[-0.05, 0.05]$. That initial shape is morphed to a different shape following the displacement field of the control nodes. The dataset contains 1000 training samples and 200 test samples generated using a second-order implicit finite volume solver. The C-grid mesh with $(220 \times 50)$ quadrilateral elements is used and adapted near the airfoil, but not around the shock.

For the forward pass, the mesh point locations and signed distance functions are passed through a trained differentiable mollified GINO model of the form

$$\mathcal{G}_{m\text{GINO}} = \mathcal{G}_{m\text{GNO}}^{decoder} \circ \mathcal{G}_{\text{FNO}} \circ \mathcal{G}_{m\text{GNO}}^{encoder} \tag{25}$$

to produce an approximation of the pressure field $p$.

#### 4.5.2 Inverse Design

For the inverse problem, we parametrize the shape of the airfoil by the vertical displacements of a few spline nodes, and set the design goal to minimize the drag-lift ratio. The parametrized displacements of the spline nodes are mapped to a mesh, which is passed through the $m$GINO to obtain a pressure field, from which we can obtain the drag-lift ratio. We optimize the vertical displacement of spline nodes by differentiating through this entire procedure. A depiction of the airfoil design problem is given in Figure 13

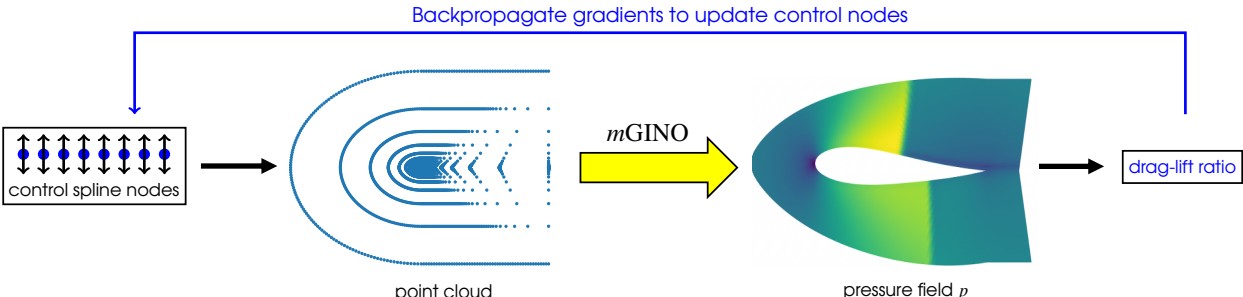

Figure 13: The airfoil design problem. Parametrized vertical displacements of control spline nodes generate a point cloud, which is passed through the differentiable $m$GINO to obtain a pressure field $p$, from which we can compute the drag-lift ratio. We update the control nodes to minimize the drag-lift ratio by differentiating through this entire procedure.

As a result of this optimization process, we obtain an airfoil design with a drag coefficient of 0.0216 and a lift coefficient of 0.2371, based on the model prediction. Using the numerical solver on this optimal design, we verify these predictions and obtain a similar drag coefficient of 0.0217 and a similar lift coefficient of 0.2532. This yields a drag-lift ratio of around 0.09, outperforming the optimal drag-lift ratio of 0.14 reported by Li et al. (2022) (drag 0.04 and lift 0.29, obtained using a Geo-FNO instead of $m$GINO).

## Discussion

We proposed $m$GNO, a fully differentiable version of GNO, to allow for the use of automatic differentiation when computing derivatives, and embedded it within GINO to learn efficiently solution operators of families of large-scale PDEs with varying geometries without data. The proposed approach circumvents the computational limitations of traditional solvers, the heavy data requirement of fully data-driven approaches, and the generalization issues of PINNs.

The use of a physics loss proved critical and was sufficient to achieve very good results with data at a very low resolution and even in the absence of data, demonstrating the data efficiency of the physics-informed paradigm. This highlights the need for efficient and accurate methods to compute physics losses, to improve data efficiency, and regularize neural operators. Autograd can compute exact derivatives in a single pass seamlessly across complex geometries and enables higher-order derivatives with minimal additional cost. However, it can be memory-intensive for deep models, possibly making it less efficient than finite differences for simpler problems. Despite these limitations, its accuracy, capability to handle complex geometries, and scalability for complicated learning tasks make it the preferred differentiation method in PINNs, and a promising approach in our physics-informed neural operator setting on complex domains.

Using Autograd instead of finite differences led to a $20\times$ reduction of the relative L2 data loss for Burgers' equation on regular grids, suggesting that the Autograd physics loss better captured the physics underlying the data. In hybrid training with noisy reference data, $m$GNO $\circ$ FNO remained robust and accurate, maintaining low PDE residuals and data loss. It also excelled at learning the lid cavity flow example, where a small number of randomly sampled points proved sufficient with Autograd to achieve excellent results. Here again, including a physics loss, potentially at a small number of randomly sampled points, proved critical. Autograd $m$GINO performed seamlessly for the Poisson and hyperelasticity equations on unstructured point clouds, while finite differences were not sufficiently accurate at the training resolution used and would need at least $9\times$ more points to compute reasonable derivatives. Autograd $m$GINO achieved a relative error 2-3 orders of magnitude lower than the Meta-PDE baselines for a comparable running time, and enjoyed speedups of $20\text{-}25\times$ and $3000\text{-}4000\times$ compared to the solver for similar accuracy on the Poisson and hyperelasticity equations. We also demonstrated with an airfoil design problem that $m$GINO can be used seamlessly for solving inverse design and shape optimization problems on complex geometries.

## Future Directions

Our framework lays the groundwork for further transformative advances in physics-informed machine learning.

Integrating techniques that have proven effective in PINNs and PINOs, such as adaptive loss reweighting and targeted sampling, could dramatically enhance training efficiency and solution accuracy, while advanced automatic differentiation strategies like the Stochastic Taylor Derivative Estimator (Shi et al., 2024) can help make high-dimensional PDEs with exact derivatives more computationally tractable. Learning data-driven mollifiers directly from PDE residuals could allow the model to adaptively tailor its smoothing properties to the structure of the underlying PDE, potentially improving both stability and generalization across regimes. Adaptive meshing further leverages the model's flexibility with arbitrary point clouds, concentrating computational effort on regions with steep gradients, singularities, or other complex dynamics, thereby accelerating training and enabling highly efficient forward and inverse design.

Data and training efficiency can be further improved by leveraging multiscale strategies. As shown throughout the paper, $m$GNO achieves high accuracy and strong data efficiency by combining low-resolution data with high-resolution physics losses. This aligns with growing recent evidence for the benefits of multiscale methods. For example, Ahamed et al. (2025) introduces a CNN training scheme across spatial resolutions, where gradual refinement maintains stable gradients and improves efficiency and generalization when direct high-resolution training would be too costly. Gal et al. (2025) propose a coarse-to-fine GNN framework using graph coarsening and multiscale gradients to reduce memory and computation without sacrificing accuracy. While we do not apply hierarchical coarsening, subsampling PDE residual points similarly reduces cost with minimal accuracy loss and suggests potential for adaptive sampling. George et al. (2024) extends the idea to FNOs with progressive training in spatial resolution and spectral modes, lowering cost, improving generalization, and producing more compact models than full-resolution training. This highlights the broader advantage of resolution adaptivity, which $m$GINO can exploit when handling point clouds, for example, by gradually increasing Fourier modes alongside resolution during training. Together, these results underscore the value of exploring multiscale extensions of $m$GNO, such as mollified kernels of varying radii, enabling simultaneous capture of short- and long-range interactions and addressing multiscale and nonlocal phenomena that challenge conventional architectures.

Together, these directions underscore the transformative potential of our approach for modeling complex, high-dimensional PDEs, opening pathways for both predictive accuracy and computational efficiency that extend well beyond current state-of-the-art methods.

## Acknowledgments

Ryan Lin is supported by the Caltech Summer Undergraduate Research Fellowships (SURF) program. Anima Anandkumar is supported in part by the Bren endowed chair, ONR (MURI grant N00014-23-1-2654), and the AI2050 Senior Fellow Program at Schmidt Sciences.

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

## A  FNO and GINO Architecture Diagrams

We first depict the Fourier Neural Operator (FNO) (Li et al., 2020a) architecture:

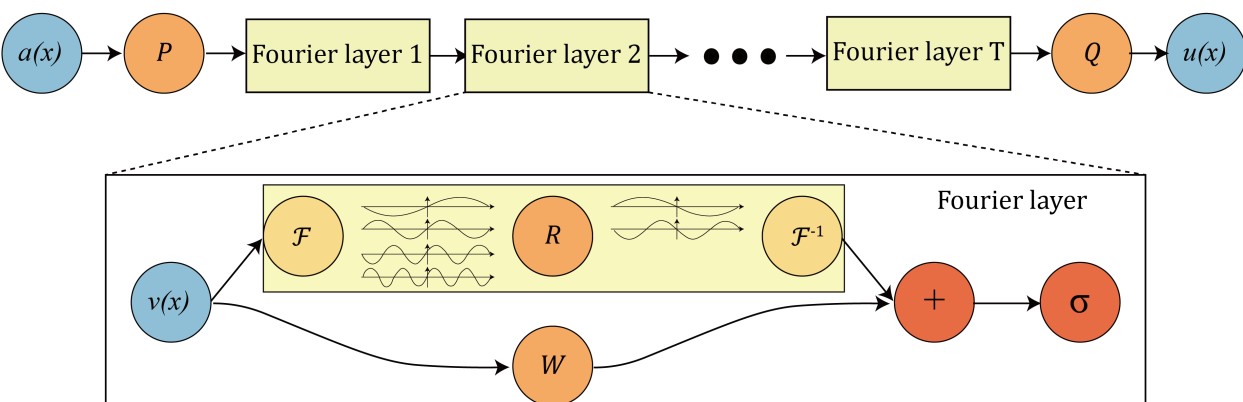

Figure 14: The Fourier Neural Operator (FNO) architecture (extracted from Li et al. (2020a)).

Next, we depict the Geometry-Informed Neural Operator (GINO) (Li et al., 2023) architecture:

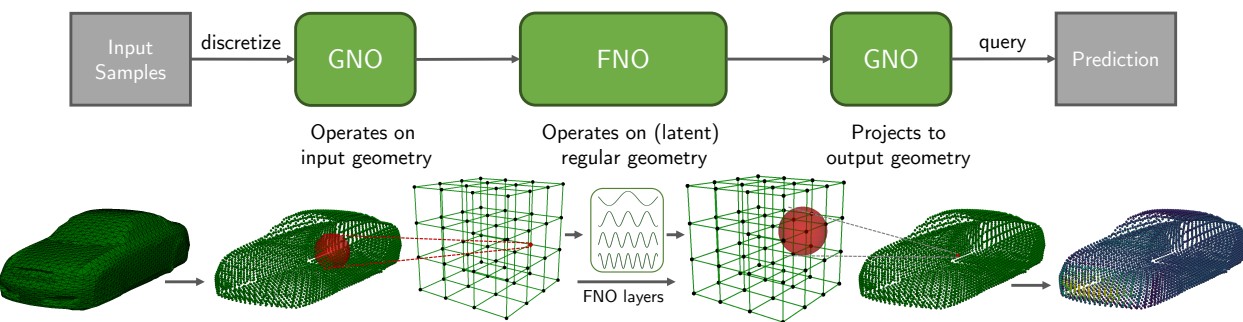

Figure 15: The Geometry-Informed Neural Operator (GINO) architecture (extracted from Li et al. (2023)).

## B  Code

The PyTorch code used for the experiments presented in this paper has been added to the open-source neural operator library from Kossaifi et al. (2025) at https://github.com/neuraloperator/neuraloperator.

## C  Choices of Hyperparameters

For all experiments, the "optimal" hyperparameters used (including weight functions and loss coefficients) were obtained by conducting a grid search on a subset of hyperparameters.

For the radius cutoff used in the weight functions of $m$GNOs, we use the same radius as for the GNO to avoid the additional cost of carrying a second neighbor search.

We use the following Mean Squared Error (MSE), Relative Squared Error, L2 error, and Relative L2 errors as metrics in our experiments:

$$\text{MSE}(y_{\text{pred}}, y_{\text{true}}) = \frac{1}{N} \sum_i^N \left( y_{\text{pred}}^{(i)} - y_{\text{true}}^{(i)} \right)^2,$$

$$\text{RELATIVESQUAREDERROR}(y_{\text{pred}}, y_{\text{true}}) = \frac{\sum_i^N \left( y_{\text{pred}}^{(i)} - y_{\text{true}}^{(i)} \right)^2}{\sum_i^N y_{\text{true}}^{(i)\,2} + \varepsilon}$$

$$\text{L2}(y_{\text{pred}}, y_{\text{true}}) = C \sqrt{\sum_i^N \left( y_{\text{pred}}^{(i)} - y_{\text{true}}^{(i)} \right)^2},$$

$$\text{RELATIVEL2}(y_{\text{pred}}, y_{\text{true}}) = \frac{C \sqrt{\sum_i^N \left( y_{\text{pred}}^{(i)} - y_{\text{true}}^{(i)} \right)^2}}{C \sqrt{\sum_i^N y_{\text{true}}^{(i)\,2}} + \varepsilon}$$

where $\varepsilon$ is a small positive number for numerical stability, and the constant $C$ is a scaling constant taking into account the measure and dimensions of the data, to ensure that the loss is averaged correctly across the spatial dimensions.

### C.1  Burgers' Equation

For Burger's equation, the input initial condition $u_0(x)$ given on a regular spatial grid in the domain $[0, 1]^2$ is first duplicated along the temporal dimension to obtain a 2D regular grid of resolution $128 \times 26$, and then passed through a 2D FNO and a mollified GNO to produce a predicted function

$$v = (\mathcal{G}_{m\text{GNO}} \circ \mathcal{G}_{\text{FNO}})(u_0)$$

approximating the solution $u(x, t)$.

For this experiment, the 2D FNO has 4 layers, each with 26 hidden channels and $(24, 24)$ Fourier modes, and we used a Tucker factorization with rank 0.6 of the weights. The $m$GNO uses the half_cos weight function with a radius of 0.1, and a 2-layer MLP with $[64, 64]$ nodes.

The resulting model has $1,019,569$ trainable parameters, and was trained in PyTorch for $10,000$ epochs using the Adam optimizer with learning rate 0.002 and weight decay $10^{-6}$, and the ReduceLROnPlateau scheduler with factor 0.9 and patience 50.

### C.2 Nonlinear Poisson Equation

For the nonlinear Poisson equation, the mesh coordinates within $[-1.4, 1.4]^2$, signed distance functions, source terms, and boundary conditions, are passed through

$$\mathcal{G}_{mGNO}^{decoder} \circ \mathcal{G}_{FNO} \circ \mathcal{G}_{GNO}^{encoder}$$

model to produce an approximation to the solution $u$.

For this experiment, the input GNO has a radius of 0.16, and a 3-layer MLP with $[256, 512, 256]$ nodes. The 2D FNO has 4 layers, each with 64 hidden channels and $(20, 20)$ Fourier modes, and acts on a latent space of resolution $64 \times 64$. The output $mGNO$ uses the half_cos weight function with a radius of 0.175, and a 3-layer MLP with $[512, 1024, 512]$ nodes.

The resulting model has $8,691,972$ trainable parameters, and was trained in PyTorch for 300 epochs using the Adam optimizer with learning rate 0.0001 and weight decay $10^{-6}$, and the ReduceLROnPlateau scheduler with factor 0.9 and patience 2. We used 7000 samples for training and 3000 samples for testing.

### C.3 Navier–Stokes Equations

For the Navier–Stokes equations, we employ a physics-informed neural operator approach that learns the mapping from spatial-temporal coordinates $(x, y, t)$ to the velocity-pressure field $(u, v, p)$. The model architecture combines a 3D FNO with a mollified GNO to produce a predicted function

$$v = (\mathcal{G}_{mGNO} \circ \mathcal{G}_{FNO})(x, y, t)$$

approximating the solution $(u(x, y, t), v(x, y, t), p(x, y, t))$.

The 3D FNO has 3 layers, each with 16 hidden channels and $(16, 16, 16)$ Fourier modes, and uses a Tucker factorization with rank 0.1 of the weights. The $mGNO$ uses the half_cos weight function with radius 0.016, a 2-layer channel MLP with $[128, 64]$ nodes. The resulting model has $431,183$ trainable parameters, and was trained in PyTorch for $15,001$ epochs using the Adam optimizer with learning rate 0.001 and the ReduceLROnPlateau scheduler with factor 0.8 and patience 200.

The training loss combines (up to) four loss components: (1) PDE residual enforcing the Navier–Stokes equations on varying numbers of randomly subsampled points from a high-resolution $256 \times 256 \times 80$ grid, (2) data loss matching velocity predictions to target data on either a varying number of points subsampled from a $64 \times 64 \times 50$ grid or a static $32 \times 32 \times 50$ grid, (3) initial condition loss enforcing zero initial velocity, and (4) boundary condition loss enforcing no-slip conditions on walls and lid-driven flow on the top boundary. Loss weighting was handled automatically and adaptively using ReLoBRaLo (Bischof & Kraus, 2021).

### C.4 Hyperelasticity Equation

For the hyperelasticity equation, the mesh coordinates within $[0, 1]^2$ and signed distance functions are passed through a

$$\mathcal{G}_{mGNO}^{decoder} \circ \mathcal{G}_{FNO} \circ \mathcal{G}_{GNO}^{encoder}$$

model to produce an approximation to the solution $u$ to the hyperelasticity equations.

For this experiment, the input GNO has a radius of 0.05625, and a 3-layer MLP with $[128, 256, 128]$ nodes. The 2D FNO has 4 layers, each with 64 hidden channels and $(20, 20)$ Fourier modes, and acts on a latent space of resolution $32 \times 32$. The output $mGNO$ uses the half_cos weight function with a radius of 0.1125, and a 3-layer MLP with $[1024, 2048, 1024]$ nodes.

The resulting model has $11,678,211$ trainable parameters, and was trained in PyTorch for 400 epochs using the Adam optimizer with learning rate 0.0001 and weight decay $10^{-6}$, and the ReduceLROnPlateau scheduler with factor 0.9 and patience 10. We used 1000 samples for training and 1000 samples for testing.

### C.5 Airfoil Inverse Design

For the airfoil design forward problem, the $220 \times 50$ mesh point locations within $[-40, 40]^2$ and signed distance functions are passed through a differentiable mollified GINO model of the form

$$\mathcal{G}_{m\text{GINO}} = \mathcal{G}_{m\text{GNO}}^{decoder} \circ \mathcal{G}_{\text{FNO}} \circ \mathcal{G}_{m\text{GNO}}^{encoder}$$

to produce an approximation of the pressure field $p$.

For this experiment, the input $m$GNO uses the half_cos weight function with a radius of 2, and a 3-layer MLP with $[128, 128, 128]$ nodes. The 2D FNO has 3 layers, each with 16 hidden channels and $(36, 36)$ Fourier modes, and acts on a latent space of resolution $64 \times 64$. The output $m$GNO uses the half_cos weight function with a radius of 6, and a 3-layer MLP with $[128, 128, 128]$ nodes.

The resulting model has $1,162,546$ trainable parameters, and was trained in PyTorch for 750 epochs using the Adam optimizer with learning rate 0.0001 and weight decay $10^{-9}$, and the ReduceLROnPlateau scheduler with factor 9 and patience 5. We used 1000 samples for training and 200 samples for testing.

## D   $m$GNO Layer Pseudocode

Here, we provide a simplified example of PyTorch pseudocode for the $m$GNO layer

$$\mathcal{G}_{m\text{GNO}}(v)(x) \coloneqq \int_{\widetilde{D}} w(x, y) \kappa(x, y) v(y) \, \mathrm{d}y,$$

with the half_cos weight function $w_{\text{half\_cos}}(x, y) = \mathbb{1}_{\mathrm{B}_r(x)}(y) \left[ 0.5 + 0.5 \cos(\pi d) \right]$, where $d = \|x - y\|^2 / r^2$.

```python
def half_cos_weight function(dists, radius=1., scale=1.):
    return scale * (0.5 * torch.cos(torch.pi * dists**2 / radius**2) + 0.5)

def mGNO_layer(
    v: Tensor[bs, n_in, codim],   # discretization of the function to transform
    y: Tensor[bs, n_in, dim],     # coordinates of the discretization of v
    x: Tensor[bs, n_out, dim],    # query locations
    delta: Tensor[bs, n_in],      # quadrature weights when approximating integral
    radius=None,                  # radius of the mollified GNO
    weight_fn=None                # weight function w,
    net: torch.nn.Module    # neural network parametrizing the kernel, e.g. a MLP
) -> Tensor[bs, n_out, codim]:

    # Kernel evaluation
    shape = [bs, n_in, n_out, dim]
    kernel_inp = [y.unsqueeze(2).expand(shape), x.unsqueeze(1).expand(shape)]
    kernel = net(torch.cat(kernel_inp, dim=-1))

    # Weighted aggregation using the quadrature weights
    output = kernel * v.unsqueeze(1) * delta.view(bs, n_in, 1, 1)

    # Mollification
    if radius is not None:
        dists = cdist(y, x)     # get distances between input and query locations
        output[dists > radius, :] = 0   # give weight 0 outside ball of radius r
        if weight_fn is not None:
            # Apply the weight function
            output = output * weight_fn(dists, radius).unsqueeze(-1)

    return output.sum(dim=-2)
```

## E  Finite Differences on Point Clouds

On a regular grid, standard well-known stencil formulas are available to compute first-order and higher-order derivatives using finite differences. However, on arbitrary point clouds, the varying distances between points have to be taken into account, and a different stencil is needed for each point at which the derivatives need to be computed. We detail below one strategy to obtain these stencil formulas in 2D.

We consider the case where we have an arbitrary point 2D cloud with $N$ points, $\{(x_i, y_i)\}_{i=1}^N$, and suppose that the function values $\{f(x_i, y_i)\}_{i=1}^N$ of a function $f$ are known at these points. The goal is to approximate partial derivatives of $f$ at any other point $(\tilde{x}, \tilde{y})$ in the domain. We start with first-order derivatives, and write them as a finite difference with unknown stencil coefficients:

$$\frac{\partial f}{\partial x}(\tilde{x}, \tilde{y}) \approx \sum_{i=1}^N c_i^{(x)} f(x_i, y_i), \qquad \frac{\partial f}{\partial y}(\tilde{x}, \tilde{y}) \approx \sum_{i=1}^N c_i^{(y)} f(x_i, y_i), \tag{26}$$

To find the stencil coefficients $c_i^{(x)}$ and $c_i^{(y)}$, we enforce that the approximation holds exactly for the functions $1$, $x$, and $y$, that is, we enforce that the approximation holds true for any polynomial of degree 1 in 2D. This results in the following systems of equations for the stencil coefficients $c_i^{(x)}$ and $c_i^{(y)}$,

$$\sum_{i=1}^N c_i^{(x)} = 0, \qquad \sum_{i=1}^N c_i^{(x)}(x_i - \tilde{x}) = 1, \qquad \sum_{i=1}^N c_i^{(x)}(y_i - \tilde{y}) = 0, \tag{27}$$

and

$$\sum_{i=1}^N c_i^{(y)} = 0, \qquad \sum_{i=1}^N c_i^{(y)}(x_i - \tilde{x}) = 0, \qquad \sum_{i=1}^N c_i^{(y)}(y_i - \tilde{y}) = 1. \tag{28}$$

This can be written as

$$A\mathbf{c}^{(x)} = \mathbf{b}^{(x)}, \qquad A\mathbf{c}^{(y)} = \mathbf{b}^{(y)}, \tag{29}$$

where

$$A = \begin{bmatrix} 1 & 1 & \cdots & 1 \\ x_1 - \tilde{x} & x_2 - \tilde{x} & \cdots & x_N - \tilde{x} \\ y_1 - \tilde{y} & y_2 - \tilde{y} & \cdots & y_N - \tilde{y} \end{bmatrix}, \tag{30}$$

and

$$\mathbf{c}^{(x)} = \begin{bmatrix} c_1^{(x)} \\ \cdots \\ c_N^{(x)} \end{bmatrix}, \qquad \mathbf{c}^{(y)} = \begin{bmatrix} c_1^{(y)} \\ \cdots \\ c_N^{(y)} \end{bmatrix}, \qquad \mathbf{b}^{(x)} = \begin{bmatrix} 0 \\ 1 \\ 0 \end{bmatrix}, \qquad \mathbf{b}^{(y)} = \begin{bmatrix} 0 \\ 0 \\ 1 \end{bmatrix}. \tag{31}$$

These systems of equations can be solved using least squares:

$$\mathbf{c}^{(x)} = (A^\top A)^{-1} A^\top \mathbf{b}^{(x)}, \qquad \mathbf{c}^{(y)} = (A^\top A)^{-1} A^\top \mathbf{b}^{(y)}. \tag{32}$$

Plugging these coefficients in Equation (26) gives the desired approximation to the first-order derivatives at $f$.

We emphasize that this procedure needs to be repeated for every point $(\tilde{x}, \tilde{y})$ at which the derivatives need to be evaluated, since the location of the point affects the entries of the matrix $A$. Note that it is not necessary and not recommended to use all $N$ points to approximate the derivatives, and one should instead identify a subset of nearest neighbors from which the finite differences can be computed.

To compute higher-order derivatives, one could follow a similar strategy (which would lead to a more complicated system of equations to solve), or first evaluate first derivatives on the point cloud and repeat the above procedure by replacing $f$ by its appropriate partial derivative.

## F    Training Times of 2D $m$GNOs with Autograd and Finite Differences

We compare the training times per epoch when training 2D $m$GNOs with $\mathcal{L}_{\mathrm{Poisson}}$ using Autograd, finite differences on uniform grids, and finite differences on non-uniform (NU) grids. Figures 16 and 17 display the training times per epoch versus the latent resolution and output resolution, respectively.

As expected, we see from Figure 16 that Autograd is more expensive than finite differences on a uniform grid at fixed latent and output resolutions. We can also see, by looking at individual columns corresponding to fixed latent space resolutions (i.e. the same model architecture), how finite differences at higher output resolutions compare to Autograd at lower output resolutions. In particular, non-uniform finite differences are often more expensive than Autograd at a slightly lower resolution. Figure 17 quantifies how running time increases when the latent space resolution of the model increases.

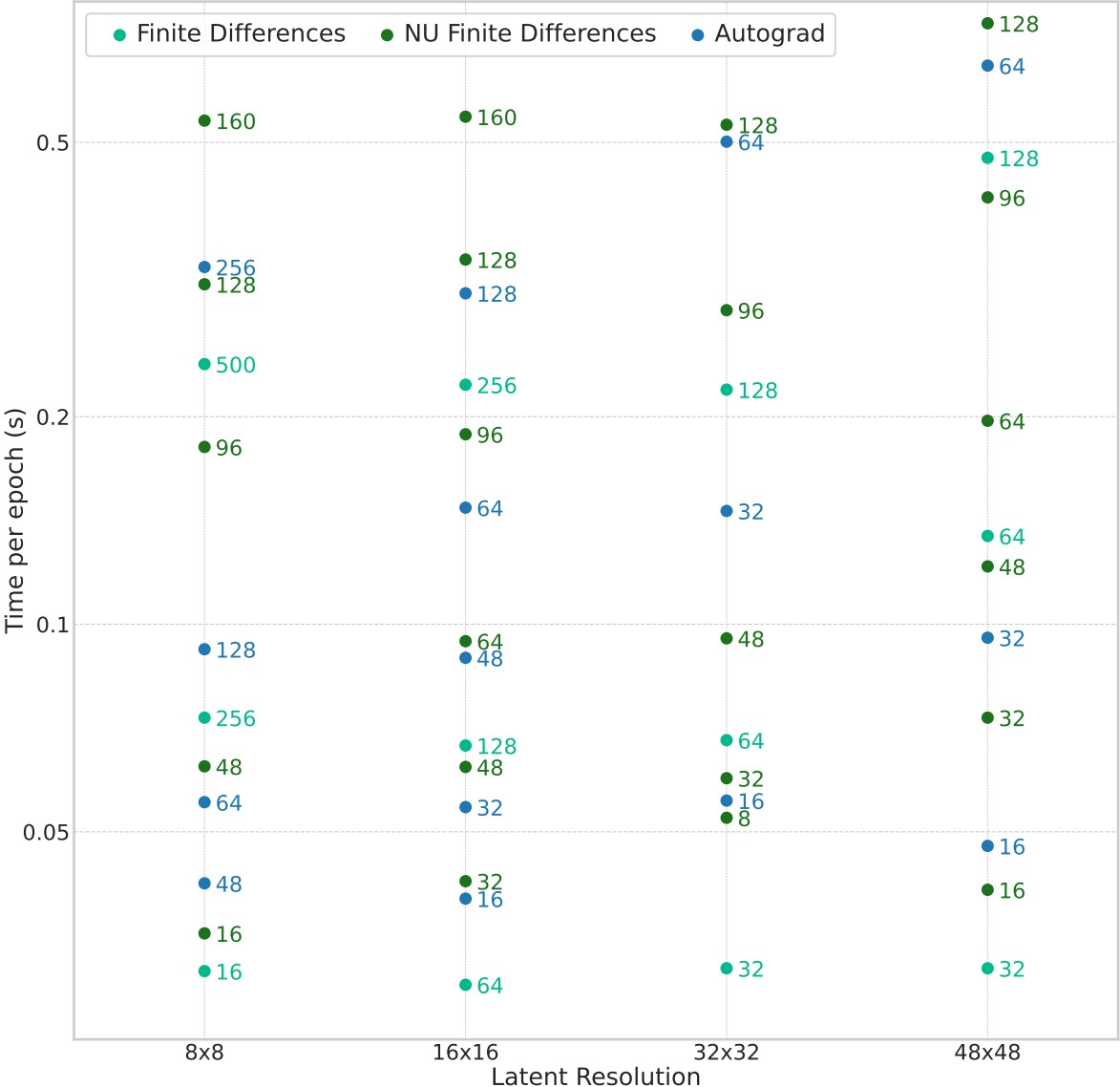

Figure 16: Time per epoch when training 2D $m$GNOs with $\mathcal{L}_{\mathrm{Poisson}}$ using Autograd, finite differences on uniform grids, and finite differences on non-uniform (NU) grids. We display the training times per epoch at different latent space resolutions *(x-axis)* and different output resolutions *(the numbers next to the points denote the number of points along each dimension)*.

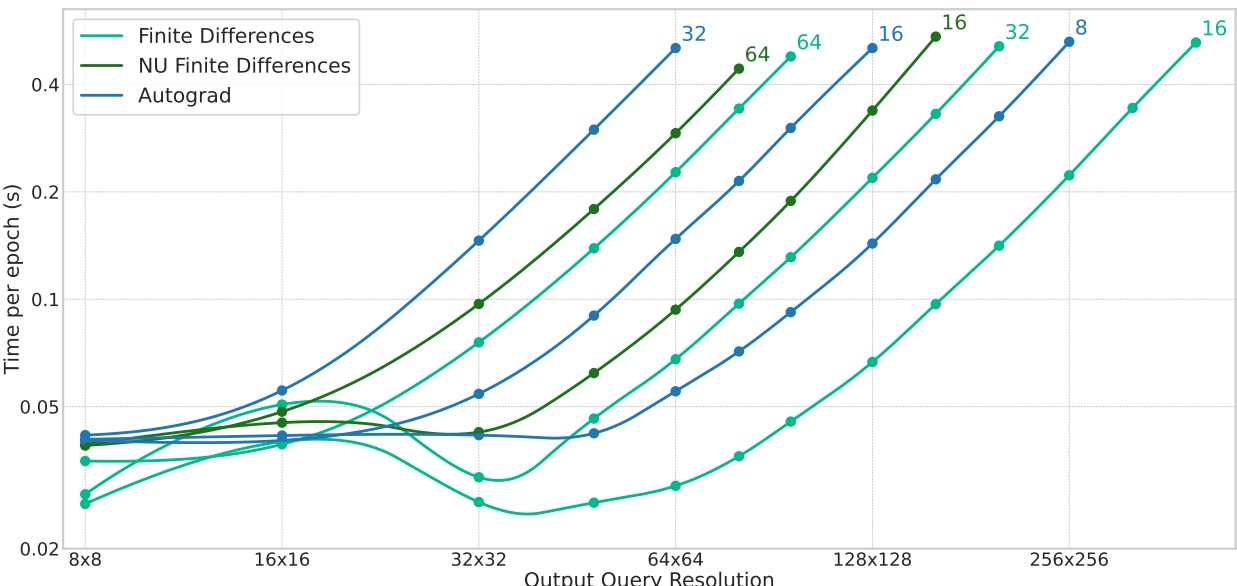

Figure 17: Time per epoch when training 2D $m$GNOs with $\mathcal{L}_{\text{Poisson}}$ using Autograd, finite differences on uniform grids, and finite differences on non-uniform (NU) grids. We display the training times per epoch at different output resolutions *(x-axis)* and different latent space resolutions *(the numbers at the end of the curves denote the number of points along each dimension)*.

## G   Automatic Differentiation with Non-Differentiable Components

In our experiments, the entire $m$GNO and $m$GINO architectures are designed such that all of their components are differentiable, ensuring that derivatives of any order can be computed correctly and accurately using automatic differentiation. For instance, we avoid non-differentiable activation functions such as ReLU (Rectified Linear Unit), which introduce discontinuities in their derivatives, and instead use smooth alternatives like GeLU (Gaussian Error Linear Unit). All the other operations in the network, including the Fourier layers, pointwise linear neural networks, and pointwise linear operators, are also implemented in a differentiable manner to avoid issues related to non-differentiability.

While automatic differentiation is naturally suited to smooth functions, it can accommodate non-differentiable components. For example, ReLU is non-differentiable at 0, yet Autograd computes exact gradients on each side and assigns a subgradient at 0 (e.g. 0 or 1). This works well in practice since exact zero inputs are rare and subgradient-based optimization remains effective. Other strategies exist for non-smooth components. Clarke subderivatives formalize generalized derivatives at non-smooth points. Proximal methods split updates into a smooth gradient step and a proximal mapping for the non-smooth part. Randomized smoothing adds small perturbations to inputs and averages outputs. Straight Through Estimators use the true non-differentiable function in the forward pass while replacing its gradient in the backward pass with a differentiable surrogate, such as a smoothed function or the identity.

Physics-informed approaches use automatic differentiation to compute derivatives of network outputs with respect to input coordinates. While automatic differentiation can handle non-differentiable components, they can be problematic for physics-informed losses. Non-smoothness can create regions where derivatives are undefined or discontinuous, and these issues can propagate through the chain rule when higher-order derivatives are needed for PDE residuals. The result can be abrupt residual changes, numerical instability, and convergence to suboptimal solutions. Differentiable components are therefore preferable since they provide continuous derivatives, ensuring smooth residuals that better capture the physics. This reduces numerical artifacts, improves gradient quality, and supports stable optimization, leading to faster convergence and more accurate solutions, with greater reliability.

