# OpenReview forum: "Enabling Automatic Differentiation with Mollified Graph Neural Operators"
_TMLR — Accepted by TMLR_

### Review · Reviewer_VJVU · 2025-07-03

**Summary Of Contributions:**

The authors substitute the original non-differentiable indicator function in the GNO kernel integral with a smooth, differentiable weighting function. This refinement makes the entire GNO operator differentiable end-to-end, enabling automatic differentiation to compute exact spatial derivatives at arbitrary query points. As a result, physics-informed loss terms based on the governing PDE can be incorporated seamlessly during training.

**Audience:**

Yes

**Claims And Evidence:**

No

**Requested Changes:**

- **Baseline selection is insufficient.** The comparison set omits several strong physics‐informed solvers. In particular, conventional PDE solvers such as Transolver [1] augmented with finite-difference PDE losses should be included as baselines; without them, it is hard to gauge the true benefit of the proposed method.
- **Alternative avenues for incorporating equation information are not explored or discussed.** Beyond loss-based enforcement, recent approaches inject physics via conditional architectures (e.g., Unisolver [2], which conditions on all available equation components) or by casting the PDE itself as a computation graph (e.g., PDEformer [3]). A thorough empirical and conceptual comparison against such strategies is necessary for a fair assessment.
- **Scalability to higher-dimensional, time-dependent problems remains unclear.** It is not evident how the current formulation extends to 2D + time PDEs—for instance, the classic Navier–Stokes dataset used in the original FNO paper [4], where a model ingests the first ten frames of a fluid sequence and predicts the next ten. Concrete guidance on applying mGNO in this setting is missing.
- **Real-world validation is lacking.** Evaluating the model on at least one realistic, noisy dataset would help demonstrate robustness and practical value beyond synthetic benchmarks.

[1] Transolver: A Fast Transformer Solver for PDEs on General Geometries

[2] Unisolver: PDE-Conditional Transformers Are Universal PDE Solvers

[3] PDEformer: Towards a Foundation Model for One-Dimensional Partial Differential Equations

[4] Fourier Neural Operator for Parametric Partial Differential Equations

###

**Strengths And Weaknesses:**

Strengths:

- The paper addresses the increasingly discussed topic of combining data-driven and physics-driven approaches; this alignment with current research trends is noted.
- Replacing the non-smooth indicator with a smooth weighting (mollifier) function is a standard idea from functional analysis and does offer a clean way to make the kernel differentiable.
- The airfoil inverse-design task is well chosen and may interest both the CFD and machine-learning communities.

Weaknesses:

- Beyond substituting a predefined smooth weight for the indicator function, the implementation changes are minimal. The core algorithmic steps remain essentially the same as existing GNO formulations, limiting the incremental contribution.

---

> ### Author Response · Authors · 2025-08-16
> **Response to Reviewer VJVU**
>
> We thank the reviewer for their insightful comments. We are pleased that the reviewer recognizes the timely nature of our work, the clean and effective approach of using a smooth mollifier function, and the relevance of the airfoil inverse-design task to both the CFD and machine learning communities.
>
> We address comments and suggestions below.
>
> ---
> ## Architectural Contribution
>
> Although the architectural modification to GNO is minimal in implementation, the mGNO constitutes a substantial improvement. By replacing the neighborhood indicator function with a smooth, compactly supported weight function, mGNO unlocks capabilities that were previously either impossible with GNO or prohibitively costly with alternative numerical differentiation schemes. This single change renders the entire operator fully differentiable, making it compatible with automatic differentiation for accurate derivative computation on arbitrary domains. Unlike finite differences, which can require high-resolution meshes to be reliable and can be especially unreliable on arbitrary point clouds, automatic differentiation with mGNO yields derivatives that are exact with respect to the computation graph and remain robust across complex geometries. Crucially, the benefits of differentiability extend far beyond training: mGNO enables end-to-end gradient-based workflows for inverse problems, design optimization, and other tasks that require backpropagation through complex geometries, all while retaining the geometric flexibility and scalability of the original GNO.
>
> ---
> ## Baselines and Alternative Approaches
> We will expand our discussion of baselines in the revised paper with the ones suggested by the reviewer, and in particular discuss how these are based on transformer architectures as opposed to our graph-based approach. We will also add a comparison to Transolver augmented with finite-difference PDE losses for the Burgers equation.
>
> ---
> ## Scalability
> Regarding time-dependence, the Burgers' equation experiment already deals with a time-dependent PDE. The input is a 1D spatial grid duplicated along the temporal dimension to create a 2D grid, which is then processed by a 2D FNO and mGNO. This approach is general and can be applied to other time-dependent problems. As suggested, we are adding a new experiment where we test our architecture on a higher-dimensional time-dependent problem, namely, a 2D+time Navier-Stokes equation (lid cavity flow from the PINO paper). We will compare the use of finite-differences and automatic differentiation. We already have good first results, but are now conducting a more systematic ablation study to investigate the effect of the resolution at which the data and physics losses are evaluated with both derivative computation approach.
>
> ---
> ## Real-World Validation
> We believe our paper already includes a strong real-world validation example: the airfoil inverse design problem, where the goal is to optimize the shape of an airfoil to minimize the drag-to-lift ratio. This experiment and the results obtained serve as a demonstration of the practical value and robustness of our method on a realistic engineering problem.
> Regarding noisy data, we are adding an ablation study that examines the effect of adding varying levels of noise to the reference data when training with a hybrid loss function that combines physics loss and data loss, and will compare the results obtained with autograd and finite differences.

---

### Review · Reviewer_eweb · 2025-07-21

**Summary Of Contributions:**

Enabling Automatic Differentiation with Mollified Graph Neural Operators introduces mGNO, a smoothed version of the Graph Neural Operator whose differentiable kernel weights let Autograd deliver exact spatial derivatives on irregular meshes and point clouds. Placing mGNO inside the Geometry‑Informed Neural Operator gives mGINO, enabling training with high‑resolution physics losses and little (or no) data while preserving differentiability across complex geometries. In experiments, Autograd‑driven mGINO cut Burgers‑equation data error by 20 × versus finite‑difference PINO, and on nonlinear Poisson and hyperelasticity tests, it achieved 2–3 orders‑of‑magnitude lower error while running 20~4 000 × faster than Meta‑PDE baselines and finite‑element solvers. The fully differentiable pipeline also supports inverse problems: optimizing an airfoil through mGINO produced a drag‑to‑lift ratio of 0.09, surpassing earlier neural‑operator designs. Overall, the paper shows that mollifying the GNO kernel unlocks exact gradients, substantial accuracy gains, major speed‑ups, and strong data efficiency for physics‑informed neural operators, extending their reach to complex geometries and design optimization tasks.

**Audience:**

Yes

**Broader Impact Concerns:**

Beyond academic PDE benchmarks, the airfoil inverse‑design experiment underscores the method’s practical relevance.
Leveraging fully differentiable aerodynamics, mGINO optimizes the wing profile, highlighting its potential for industrial‑scale design workflows that would otherwise depend on expensive CFD loops, matching or beating domain‑specific surrogate optimizers without additional heuristics.
This showcases a valuable new capability unlocked by the technique.

**Claims And Evidence:**

Yes

**Requested Changes:**

The single paragraph on future work (p. 17) is brief and not critical.
Ideas like adaptive or hybrid losses appear tangential to the core contribution of smoothing the kernel weights in Eq. 6.
A richer outlook, such as learning data‑driven mollifiers, coupling with adaptive meshing, or extending to multi‑scale operators—would better demonstrate the method’s potential and improve its novelty.

**Strengths And Weaknesses:**

# Strengths

1. Clear, well‑motivated problem statement. The authors pinpoint a concrete bottleneck in existing neural operators: the binary neighbor indicator breaks automatic differentiation, preventing the use of gradient‑based physics losses and downstream design tasks.  By targeting this pain‑point the paper delivers a focused, easily understood contribution that resonates with both ML and computational‑physics audiences.

2. Comprehensive experimental coverageThe evaluation spans:  (i) 1‑D viscous Burgers, (ii) 2‑D nonlinear Poisson, (iii) 3‑D hyper‑elasticity, and (iv) a real‑world NACA air‑foil inverse‑design task.  The breadth demonstrates robustness across equation families, dimensionalities, and mesh irregularities, while the air‑foil case illustrates real engineering impact.

# Weakness

Incremental methodological novelty. The core idea—smooth a discontinuous indicator—can be expressed in a single line of code.  Reviewers looking for deeper algorithmic insights or theory may find the contribution thin relative to the empirical section.

---

> ### Author Response · Authors · 2025-08-16
> **Response to Reviewer web**
>
> We sincerely appreciate your feedback and enthusiasm regarding the practical industrial impact of our work. Your recognition of it as a significant advancement for industrial design workflows is truly encouraging.
>
> We address your comments and suggestions below.
>
> ---
>
> ## Architectural Contribution
>
> Although the architectural modification to GNO is minimal in implementation, the mGNO constitutes a substantial improvement. By replacing the neighborhood indicator function with a smooth, compactly supported weight function, mGNO unlocks capabilities that were previously either impossible with GNO or prohibitively costly with alternative numerical differentiation schemes. This single change renders the entire operator fully differentiable, making it compatible with automatic differentiation for accurate derivative computation on arbitrary domains. Unlike finite differences, which can require high-resolution meshes to be reliable and can be especially unreliable on arbitrary point clouds, automatic differentiation with mGNO yields derivatives that are exact with respect to the computation graph and remain robust across complex geometries. Crucially, the benefits of differentiability extend far beyond training: mGNO enables end-to-end gradient-based workflows for inverse problems, design optimization, and other tasks that require backpropagation through complex geometries, all while retaining the geometric flexibility and scalability of the original GNO.
>
> ---
>
> ## Future Work
> We agree that the discussion on future work should be more substantive and thank you for the valuable suggestions. We will expand the final paragraph of the discussion to present a richer research outlook, outlining the following directions:
>
> - **Learning data-driven mollifiers:** Instead of relying on predefined weight functions, we could learn an optimal mollifier directly from data. This would enable the model to adapt its smoothing properties to the specific PDE characteristics, potentially improving performance.
>
> - **Leveraging adaptive meshing:** Since our method can handle arbitrary point clouds, adaptive meshing could be leveraged to accelerate training or enhance inverse design by progressively refining shapes during optimization.
>
> - **Extending to multi-scale operators:** We could combine mGNO layers with mollified kernels of varying scales to capture both short- and long-range interactions more effectively.
>
> - **Higher-order automatic differentiation:** As already discussed in the current version, leveraging advanced automatic differentiation techniques could help amortize computational costs and improve efficiency.

---

### Review · Reviewer_r2Xe · 2025-08-11

**Summary Of Contributions:**

The paper proposes mollified Graph Neural Operators (mGNO) and a mollified GINO (mGINO) that replace the hard neighborhood indicator with smooth, compactly supported weights, enabling automatic differentiation (AD) of physics losses on irregular geometries and point clouds. This allows training with PDE residuals at arbitrary query points and supports inverse design by backpropagating through geometry.

Experiments span 1D Burgers (on grids), nonlinear Poisson and hyperelasticity (on unstructured point clouds), and an airfoil optimization. Across tasks, AD with mGNO is reported to reduce training error relative to finite differences, outperform a meta-learning baseline on irregular domains at comparable runtime, and achieve notable speedups versus a numerical solver at similar accuracy; the airfoil design is validated with a high-fidelity solver.

**Audience:**

Yes

**Broader Impact Concerns:**

I am not aware of negative impacts of this work.

**Claims And Evidence:**

Yes

**Requested Changes:**

I think that overall the paper is very interesting and well sectioned and written. There are some changes that need to be done before acceptance:

1.  Clarify derivative claims, for example by replacing “exact derivatives/gradients” with phrasing such as “AD-exact with respect to the differentiable computation graph,” and briefly discuss discretization effects, support-boundary/neighbor changes, and their practical impact on stability and smoothness.

2. Address non-smooth components. Clarify how you handle ReLU/max/clip and discrete neighbor updates: which derivative notion is used at kinks, whether smooth activations were considered for higher-order physics losses, and any observed differences (a small ablation would suffice).

3. Differentiate from “physics-informed approaches without physics losses.” In that paragraph, explicitly state how your method differs from architectures that encode physics via constraints/projections/operator design rather than explicit PDE residuals. Note when each route is preferable (e.g., memory/compute vs flexibility on irregular domains).

4. Weight-functions context. Before listing specific weights, add 1–2 sentences summarizing purpose (mollify hard neighborhoods for stable AD), key properties (smoothness, compact support, controllable locality), and default recommendations; note how radius is chosen (e.g., target neighbor count).

5. For completeness, please add the initial conditions for Burgers equation solution in Figure 4.

6. Regarding subsampling and data efficiency, please discuss recent training strategies such as:

- Multiscale Training of Convolutional Neural Networks

- Towards efficient training of graph neural networks: A multiscale approach

That also consider the data efficiency and time efficiency of training neural networks, with some overlap to the considerations in this paper.

7. Since your method shows better approximation of derivatives, it would be interesting to quantify its stability or robustness. Can the authors perform an experiment with noisy inputs to show the performance of their method vs finite differences ?

**Strengths And Weaknesses:**

Strengths:

1. Mollifying the neighborhood kernel is a clean fix that makes GNO/GINO compatible with AD-based physics losses on arbitrary domains.
2. There are clear improvements over finite differences for training with physics losses, strong point-cloud results, and a credible inverse-design demo with solver verification.
3.  The ablations on radius/weights, subsampling of physics points, and notes on aggregation offer actionable tuning advice.

Weaknesses:

1. AD is exact for the computation graph, but gradients can still be affected by discretization, neighborhood changes at the support boundary, and non-smooth components. The phrasing should be carefully revised to be clear.

2. Missing discussion on non-smooth operations: many operator stacks use ReLU/max/clipping and discrete neighbor updates. Claims about higher-order derivatives and smoothness need to address these cases explicitly. It might be that this is already taken care of by the Authors, by I could not understand this from the paper.

3. Missing aggregation and radius sensitivity discussion: the paper touches on sum vs mean and shows a weight ablation, but a broader view of aggregation schemes and a clearer accuracy–cost frontier versus radius/neighbor count would help to improve its quality.

4. The “Weight functions” paragraph drops in abruptly without re-stating purpose and design requirements.

---

> ### Author Response · Authors · 2025-08-16
> **Response to Reviewer r2Xe (1/2)**
>
> Thank you for your review, we appreciate your positive assessment of our work, particularly the strength of our experimental results and the practical advice on tuning.
>
> We address your comments and suggestions below.
>
> ---
>
> ## 1. Derivative Claims
> You are right to point out that the term "exact" requires careful qualification. Automatic differentiation (AD) computes gradients exactly with respect to the implemented differentiable computation graph, yielding derivatives accurate up to machine precision. However, since this graph typically encodes a discretized approximation of a continuous PDE, both the computed loss and its gradients reflect the fidelity of that discretization, which depends on the number and placement of evaluation points. Despite this inherent approximation, our results show that the approach remains robust, maintaining strong performance even when the PDE loss is evaluated on randomly subsampled points. We will revise the manuscript to make this clarification and emphasize the importance of discretization choices on gradient accuracy, as well as highlight the demonstrated robustness of our method under varying sampling strategies.
>
> ---
>
> ## 2. Address Non-Smooth Components
> This is an important point, and we sincerely thank the reviewer for highlighting it. Our model uses GeLU, which is a smooth activation function and is differentiable everywhere. Thus, it does not introduce the non-differentiable “kinks” that functions like ReLU do. This means our model can compute exact higher-order derivatives without the issues associated with subgradients or undefined derivatives at specific points. We will emphasize this more clearly in the paper, and also note explicitly that all other components in our architecture are now differentiable.
>
> ---
>
> ## 3. Differentiate from "Physics-Informed Approaches without Physics Losses"
> We will expand the discussion in this section (Section 2.2) to better highlight the distinction.
> PINNs, PINO, mGNO, and mGINO explicitly incorporate physical laws by including physics-based loss terms during training. This approach treats the governing equations as regularizers, providing greater flexibility and broad applicability across PDEs. Only knowledge of the governing equations is required.
>
> In contrast, physics-informed methods without physics losses incorporate physical laws directly into the model architecture, for example, through projection layers or carefully chosen basis functions. These approaches are especially useful when the system is well-understood and physical constraints can be explicitly encoded, resulting in guaranteed compliance with the underlying physics. They also generally demand less computational overhead. However, such methods are usually tailored to very specific PDE structures and are limited to dynamical systems with well-characterized or well-understood solutions.
>
> A hybrid approach is also possible. For example, in incompressible Navier Stokes problems, a physics loss can be used to penalize deviations from the momentum equation while the divergence-free condition can be enforced directly within the model as in the references listed in the paper.
>
> ---
>
> ## 4. Weight-Functions Context
> Thank you for your suggestion, we agree that adding a few sentences on this would improve clarity. We will revise the introduction to this paragraph as suggested. We will also provide there some guidance on how the radius can be tuned (and connect it to the ablation study of Section 4.2.6 on the choice of radius) and on the choice of aggregation scheme (and connect it to the discussion of Section 4.2.3).
>
> ---
>
> ## 5. Add Initial Conditions for Burgers' Equation in Figure 4
> For Burgers’ equation, the initial conditions are drawn from a Gaussian process (see Section 4.1.1). We will add a plot of different examples of initial conditions.

---

> > ### Author Response · Authors · 2025-08-16
> > **Response to Reviewer r2Xe (2/2)**
> >
> > ---
> >
> > ## 6. Multiscale Training Strategies
> > We appreciate the reviewer’s suggestion to discuss recent multiscale training strategies and how they relate to the efficiency achieved in our work. Our work already demonstrates a form of data efficiency by showing that our method can achieve high accuracy by supplementing low-resolution data with high-resolution physics losses. We will update the manuscript to situate mGNO within the broader literature on data and training efficiency in this field.
> >
> > - ***Multiscale Training of CNNs (Ahamed et al., 2025)*** proposes a multi-resolution optimization scheme where CNNs are trained across multiple spatial scales. By progressively refining resolution while maintaining stable gradients, the method improves both computational efficiency and generalization, especially when high-resolution training from scratch would be prohibitively costly. This idea is conceptually related to our ability to compute physics losses at arbitrary resolutions. In mGNO, smooth mollifier kernels ensure stable AD-derived derivatives across resolutions, enabling us to train effectively at coarser physics-loss resolutions while still achieving accurate high-resolution predictions, without retraining at each scale.
> >
> > - ***Towards Efficient Training of Graph Neural Networks: A Multiscale Approach (Gal et al., 2025)*** develops a coarse-to-fine GNN training framework that uses graph coarsening and multiscale gradient computation to reduce computation and memory requirements while preserving accuracy. While we do not explicitly apply hierarchical coarsening, using subsampling of PDE residual evaluation points (Sec. 4.1.3), we can significantly cut computational cost while retaining nearly identical accuracy over a wide range. In addition, an interesting future direction would be to explore the efficient use of adaptive sampling strategies with our approach.
> >
> > - ***Incremental Spatial and Spectral Learning of Neural Operators for Solving Large-Scale PDEs (George et al. 2024)*** proposes a progressive training strategy for FNOs in which both the spatial resolution and the number of retained Fourier modes are increased gradually over the course of training. This approach reduces computational cost, improves generalization, and produces more compact models compared to training directly at full resolution. It illustrates the broader advantage of using resolution adaptivity to balance accuracy and efficiency in neural operator learning, a principle that mGINO can also exploit when learning from point clouds of varying sizes. A similar progressive approach could be applied to the FNO component within mGINO by gradually increasing the number of retained Fourier modes during training, alongside varying the resolution.
> >
> > ---
> >
> > ## 7. Experiment with Noisy Inputs
> >
> > We thank the reviewer for this suggestion. If we perturb slightly the initial conditions, we are effectively setting up a different PDE instance. Then, any model trained with a physics-based loss would be learning that new instance. As a result, this would really demonstrate more about the sensitivity of the PDE with respect to the initial conditions, rather than the sensitivity of the model or derivative computation scheme. In our setting, it is given that the initial conditions and the governing equations are known and fixed. In alignment with the suggestion of considering robustness to noise, we are adding an ablation study that examines the effect of adding varying levels of noise to the reference data when training with a hybrid loss function that combines physics loss and data loss, and will compare the results obtained with autograd and finite differences.

---

> > ### Comment · Reviewer_r2Xe · 2025-08-17
> >
> > Thank you for the responses. In point 2 (non smooth components) I was asking - can you extend your work and discussions to non-smooth components? I understand that in your current experiments you only used smooth and differentiable components, but I am interested in knowing how would it work in cases where one uses activations like ReLU for example?
> >
> >
> > I am happy with the rest of your responses, and I am looking forward to seeing the new experiments.

---

> > > ### Author Response · Authors · 2025-08-22
> > > **Discussion of non-smooth components**
> > >
> > > Thank you for the clarification. While automatic differentiation is naturally suited to smooth functions, it can accommodate non-differentiable components via generalized derivatives and subgradients. For instance, for ReLU (non-differentiable at 0), autograd provides exact gradient computation on each side of 0, and defines the derivative at 0 as a subgradient, often arbitrarily choosing 0 or 1. This works reasonably well in standard neural network training because exact zero inputs are rare, and subgradient-based optimization remains effective. Several additional strategies exist to handle non-smooth components. Clarke subderivatives provide a more formal perspective, defining a set of valid generalized derivatives at non-smooth points and explaining why automatic differentiation can still yield meaningful gradients even when functions are not fully differentiable. Proximal methods are also widely applied by separating updates into a gradient step on the smooth part and a proximal mapping for the non-smooth part. Randomized smoothing approximates by adding small perturbations to inputs and averaging outputs. Alternatively, one can use Straight Through Estimators. In this case, it would consist in using the actual (non-differentiable) function during the forward pass. In the backwards pass, however, instead of using that non-differentiable function's gradient, we replace it by a differentiable surrogate (e.g. a smoothed version, or even the identity).
> > >
> > > Now, physics-informed approaches rely on automatic differentiation to compute derivatives of network outputs with respect to input coordinates. While automatic differentiation can handle non-differentiable components using the approaches mentioned earlier, this is not ideal for physics-informed losses. Non-smoothness introduces points or regions where the derivative is undefined or discontinuous, which can propagate through the chain rule when computing higher-order derivatives required for PDE residuals. This can result in abrupt changes in the residuals, creating numerical instability and making gradient-based optimization less reliable and possible convergence to suboptimal solutions. Differentiable components are therefore highly preferable in this context as they provide well-defined continuous derivatives which ensures that PDE residuals are smooth and accurately represent the underlying physics. Differentiable components reduce numerical artifacts, improve gradient signal quality, and facilitate stable and efficient optimization, ultimately resulting in faster convergence, more precise solutions, and greater reliability.

---

> > > > ### Comment · Reviewer_r2Xe · 2025-08-23
> > > >
> > > > I thank the authors for the clarifications and for the added experiments in the revised paper. I have no further questions.

---

### Author Response · Authors · 2025-08-16
**Response to the reviewers**

We sincerely thank all the reviewers for their thoughtful feedback and constructive suggestions.

We will address each comment individually and update the manuscript, including the PDF on OpenReview, within the next few days.


The main revisions we plan to incorporate based on the reviewers’ input are summarized below:

---

### Clarifications and additional discussions
- **Derivative Claims**: Clarify “exact” gradients in automatic differentiation
- **Non-Smooth Components**: Emphasize all model components are differentiable
- **Physics Loss vs. Architectural Physics**: Highlight distinction between physics-loss-based methods (PINNs, PINO) and those embedding physics directly
- **Weight Functions**: Expand motivation and details of the section
- **Burgers’ Equation initial conditions**: Add plot showing examples of ICs
- **Multiscale Training**: Situate work in context of multi-resolution/progressive training literature, and connect to data efficiency
- **Baselines & Alternatives**: Expand baseline discussion, in particular transformer-based methods
- **Future work**: We will extend the paragraph on future work

---
### Further Experiments
- **Baseline**:  Add comparison to Transolver with finite-difference PDE losses for Burgers’ equation.
- **Scalability**: Add experiment on 2D+time Navier-Stokes equations
- **Noisy data**:   Add an ablation study to test the effect of noisy data when training with hybrid loss, for Burgers’ equation

---

> ### Author Response · Authors · 2025-08-22
> **The revised manuscript has been uploaded to OpenReview**
>
> We would like to thank the reviewers once again for their valuable feedback.
>
>  The main revisions suggested have been incorporated into the manuscript, and the revised version has been uploaded to OpenReview.

---

### Decision · Action_Editor_SBPD · 2025-09-24

**Recommendation:** Accept as is

**Audience:**

Yes

**Audience Explanation:**

This paper falls into the neural operator field, a subfield of the more broad scientific machine learning discipline. It will attract researchers and practitioners interested in AI for Science.

**Claims And Evidence:**

Yes

**Claims Explanation:**

The submission substitutes the non-differentiable indicator function in the Graph Neural Operator (GNO) kernel integral with a smooth, differentiable weighting function. This enables differentiable GNO operator with automatic differentiation to compute exact spatial derivatives at arbitrary query points. This contribution is meaningful in the physics-informed ML area, with all the arguments made in the paper well grounded as acknowledged by all reviewers.